# Dub3 inhibition suppresses breast cancer invasion and metastasis by promoting Snail1 degradation

Yadi Wu[1,2], Yu Wang[1,2], Yiwei Lin[2,3], Yajuan Liu[2,3], Yifan Wang[2,3], Jianhang Jia[2,3], Puja Singh[4], Young-In Chi[4], Chi Wang[2,5], Chenfang Dong[6], Wei Li[7], Min Tao[7], Dana Napier[2,8], Qiuying Shi[2,8], Jiong Deng[9], B. Mark Evers[2,10] & Binhua P. Zhou[2,3,11]

Snail1, a key transcription factor of epithelial–mesenchymal transition (EMT), is subjected to ubiquitination and degradation, but the mechanism by which Snail1 is stabilized in tumours remains unclear. We identify Dub3 as a bona fide Snail1 deubiquitinase, which interacts with and stabilizes Snail1. Dub3 is overexpressed in breast cancer; knockdown of Dub3 resulted in Snail1 destabilization, suppressed EMT and decreased tumour cell migration, invasion, and metastasis. These effects are rescued by ectopic Snail1 expression. IL-6 also stabilizes Snail1 by inducing Dub3 expression, the specific inhibitor WP1130 binds to Dub3 and inhibits the Dub3-mediating Snail1 stabilization *in vitro* and *in vivo*. Our study reveals a critical Dub3–Snail1 signalling axis in EMT and metastasis, and provides an effective therapeutic approach against breast cancer.

[1] Department of Pharmacology & Nutritional Sciences, The University of Kentucky, College of Medicine, Lexington, Kentucky 40506, USA. [2] Markey Cancer Center, The University of Kentucky, College of Medicine, Lexington, Kentucky 40506, USA. [3] Department of Molecular and Cellular Biochemistry, The University of Kentucky, College of Medicine, Lexington, Kentucky 40506, USA. [4] The Hormel Institute, University of Minnesota, Austin, Minnesota 55912, USA. [5] Department of Biostatistics, The University of Kentucky, College of Medicine, Lexington, Kentucky 40506, USA. [6] Department of Pathology and Pathophysiology, Zhejiang University School of Medicine, Zhejiang 310058, China. [7] Department of Oncology, The First Affiliated Hospital of Soochow University, PREMED Key Laboratory for Precision Medicine, Soochow University, Suzhou 215006, China. [8] Department of Pathology, The University of Kentucky, College of Medicine, Lexington, Kentucky 40506, USA. [9] Key Laboratory of Cell Differentiation and Apoptosis of Chinese Minister of Education, Shanghai Jiao Tong University School of Medicine, Shanghai 200025, China. [10] Department of Surgery, the University of Kentucky, College of Medicine, Lexington, Kentucky 40506, USA. [11] State Key Laboratory of Oncology in South China, Sun Yat-sen University Cancer Center, Collaborative Innovation Center for Cancer Medicine, Guangzhou 510060, China. Correspondence and requests for materials should be addressed to Y.W. (email: yadi.wu@uky.edu) or to B.P.Z. (email: ZhouBh@sysucc.org.cn or peter.zhou@uky.edu).

Approximately 90% of cancer death are caused by metastasis[1], which is an exceedingly complex process involving tumour cell motility, intravasation, circulation in the blood or lymph system, extravasation and growth in new tissues and organs. The increased motility and invasive properties of metastatic tumour cells are reminiscent of events that occur during epithelial–mesenchymal transition (EMT), which is a distinctive morphogenic process that occurs during embryonic development, chronic degeneration and fibrosis of organs, and tumour invasion and metastasis[2,3]. During EMT, epithelial cells acquire fibroblast-like properties, exhibit reduced intercellular adhesion and show increased motility. Several transcription factors are associated with EMT, including the Snail1/Slug family[4], Twist[5], δEF1/ZEB1 and SIP1/ZEB2 (refs 6,7).

Snail1, a zinc-finger containing transcription factor, was identified in *Drosophila* as a suppressor of *shotgun* (an E-cadherin homologue) transcription, which controls large-scale cell movement during mesoderm formation and neural crest delamination[4]. Snail1 expression is tightly regulated during development; this regulation is often disrupted in metastatic breast cancer. Overexpression of Snail1 was found in both epithelial and endothelial cells of invasive breast cancer[8]. Snail1 expression correlates with the tumour grade and nodal metastasis for invasive ductal carcinoma[9–11] and predicts a poor outcome in patients with breast cancer[12]. Snail1 overexpression also induces resistance to apoptosis, confers tumour recurrence and generates breast cancer stem cell (CSC)-like properties[13,14]. We recently found that Snail1 induces aerobic glycolysis by repressing fructose-1,6-biphosphatase (FBP1) expression, and thus provides metabolic growth advantages to breast cancer[15].

Although several signalling pathways, such as EGF, FGF, HGF, TGFβ and Notch, can induce Snail1 transcription under different cellular contexts[16], Snail1 is a labile protein and is under constant protein ubiquitination and degradation mediated by FBXL14, β-TRCP1 or FBXO11 (refs 11,17,18). For example, phosphorylation of Snail1 by glycogen synthase kinase-3β (GSK-3β) promotes Snail1 export from the nucleus. In the cytoplasm, Snail1 undergoes a second phosphorylation by GSK-3β, which targets the protein for β-TRCP1-mediated cytoplasmic degradation. In addition, PDK1 phosphorylates Snail1 to form a Snail1–FBXO11 complex in the nucleus[17]. On the other hand, we reported that Snail1 stabilization is induced by the inflammatory cytokine TNFα through the NF-κB pathway to block Snail1 ubiquitination[19]. However, a comprehensive account of the mechanisms by which Snail1 escapes ubiquitination and degradation in breast cancer remains unknown.

Ubiquitination is a reversible process and ubiquitin moieties are removed from polypeptides by Deubiquitinases (DUBs). DUBs are classified into ubiquitin C-terminal hydrolase (UCH), ubiquitin-specific processing proteases (USP), Jab1/Pad1/MPN-domain containing metallo-enzymes (JAMM), Otu domain ubiquitin-aldehyde binding proteins (OTU) and Ataxin-3/Josephin-domain containing proteins (Ataxin-3/Josephin). Growing evidence shows that DUBs are essential for the regulation of many cellular functions including transcription, DNA repair and cell cycle progression[20]. Dub3 belongs to the USP group, and is an immediate early gene that belongs to a subfamily of cytokine-inducible DUBs[20]. Specifically, Dub3 is rapidly induced by IL-4 and IL-6 (refs 21,22). Cdc25A is a known substrate of Dub3 that promotes oncogenic transformation[23]. In agreement with this report, high Dub3 expression in mouse embryonic stem cells couples the G1/S checkpoint to pluripotency through regulation of Cdc25A (ref. 24), and depletion of Dub3 from breast cancer cells reduces proliferative potential *in vivo*. In addition to the role in breast cancer, Dub3 expression correlates with tumour progression and poor prognosis in human epithelial ovarian cancer[25]. However, these observations do not specifically explain the role of Dub3 in mediating tumour cell invasion and metastasis.

In the current study we utilize unbiased approaches to identify the specific DUB responsible for Snail1 stabilization, and identify Dub3 as a bona fide DUB of Snail1. The Dub3–Snail1 signalling axis forms a 'sensor and effector' circuitry by overlaying inflammatory stimulation to EMT and metastasis.

## Results

**Dub3 is a deubiquitinase of Snail1.** To understand the regulation of Snail1, we purified the Snail1 complexes from nuclear extracts of 20 l HeLa S3 cells expressing Flag-Snail1 (ref. 26). The immunocomplex was separated on SDS–PAGE and subjected to top-down mass spectrometry analysis. We determined that several histone methyltransferases/demethylases, such as LSD1 (ref. 26), Suv39H1 (ref. 27) and G9a (ref. 28) as well as Dub3, were associated with Snail1 (Supplementary Fig. 1a). In a parallel experiment, we performed a small interfering RNA (siRNA) library screening, which consisted of four non-overlapping siRNA targeting the 99 known or putative DUBs. This initial screen identified 11 genes that may directly or indirectly control Snail1 stability (Supplementary Fig. 1b). When these DUBs were co-expressed with Snail1 in HEK293 cells, we found that USP12, Dub3 and USP28 significantly increased Snail1 levels, similar to results obtained when cells were treated with the proteasome inhibitor MG132 (Supplementary Fig. 1b). However, only Dub3 interacted with Snail1 in the co-immunoprecipitation (IP) assay (Supplementary Fig. 1c). These two independent and unbiased analyses point to the critical role of Dub3 in the regulation of Snail1.

To further investigate the relationship of these two proteins, we co-expressed Snail1 with Dub3 in HEK293 cells. Expression of wild-type (WT) Dub3 stabilized Snail1. A Dub3 mutant, in which the catalytic cysteine had been replaced with serine (C89S, CS), showed no such effect, indicating that the enzymatic activity of Dub3 is required for Snail1 stabilization (Fig. 1a). A steady-state level of Snail1 was enhanced by increasing Dub3 expression in a dose-dependent manner (Fig. 1b). When Dub3 was co-expressed with GFP-Snail1 in HEK293 cells, we found that Dub3 stabilized and co-localized with GFP-Snail1 in nuclei (Fig. 1c). Although we did not find any correlation between Dub3 and Snail1 mRNA levels, expressions of Dub3 and Snail1 in multiple cancer cell lines, ranging from colon, prostate and breast tumours, were highly correlated (Fig. 1d). Dub3 was highly expressed in basal-like breast cancer (BLBC) cells that contain high levels of Snail1. In addition, Dub3 expression correlated with Snail1 in colon and prostate cancer cell lines, suggesting that this Dub3–Snail1 correlation is not tissue-specific. Dub3 expression also correlated with Snail1 levels in 12 cases of fresh breast tumours (Fig. 1e). These data suggest that Dub3 controls the level of Snail1 through deubiquitination to prevent degradation. Consistent with this idea, knockdown of endogenous Dub3 resulted in a rapid loss of endogenous Snail1 protein, but had no effect on mRNA levels, in MDA-MB231 and MDA-MB157 cells (Fig. 1f). The downregulation of Snail1 in Dub3-knockdown MDA-MB157 cells was restored by MG132 treatment (Fig. 1g), indicating that Dub3-knockdown facilitates the ubiquitination and degradation of Snail1.

Dub3 is evolutionarily conserved from *Drosophila* to humans[29]. Strikingly, knocked-out Dub3 expression using UAS-RNAi lines that target Dub3 in *Drosophila*, show no invagination/gastrulation, which require both EMT and stem

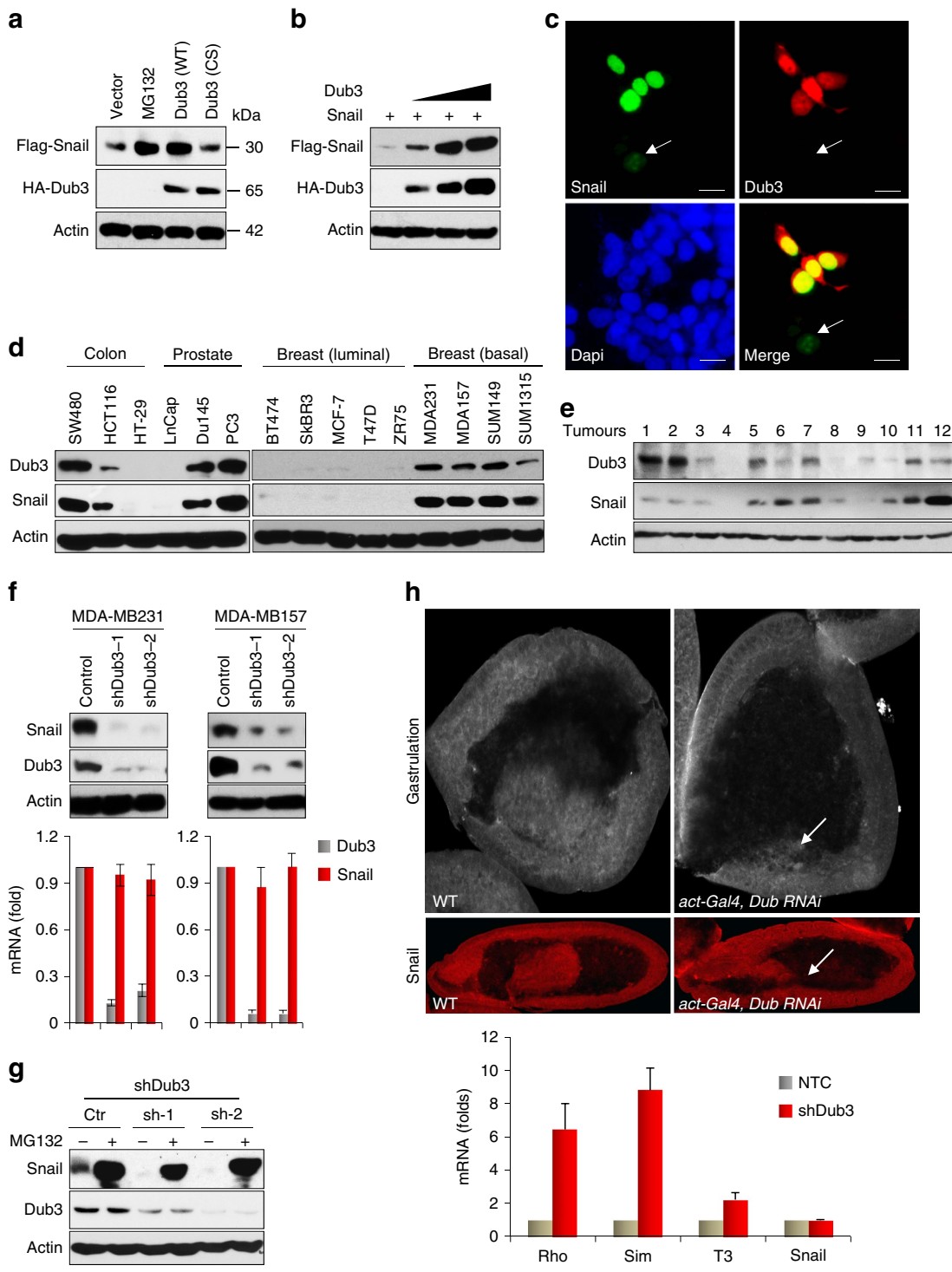

**Figure 1 | Dub3 stabilizes Snail1.** (**a**) Flag-Snail1 was co-expressed with HA-tagged Dub3 (either wild-type, WT, or catalytic inactive C89S mutant, CS) in HEK293 cells or cells were treated with MG132 for 6 h. Expression of Snail1 and Dub3 were assessed by western blot. (**b**) Flag-Snail1 was co-expressed with increasing amounts of HA-Dub3 in HEK293 cells. Lysates were subjected to analysis by western blot. (**c**) GFP-Snail1 was co-expressed with HA-Dub3 in HEK293 cells. After fixation, the cellular location of Snail1 (green) and Dub3 (red) was examined by immunofluorescent (IF) staining using anti-HA antibody and visualized by fluorescence microscopy (nuclei were stained with Dapi; blue). Arrowhead identifies a cell expressing only GFP-Snail1 but not Dub3. Scale bars, 25 μm. (**d**) The protein expression of Dub3 and Snail1 in various cancer cell lines was analysed by western blot. (**e**) Expression of Dub3 and Snail1 from 12 human breast tumours (fresh frozen) was analysed by western blot. (**f**) The protein expression of Dub3 and Snail1 from MDA-MB157 and MDA-MB231 cells stably transfected with control or two individual Dub3 shRNAs was analysed by western blot and the mRNA was detected by real-time PCR (mean ± s.e.m. in three separate experiments). (**g**) The protein expression of Dub3 and Snail1 from MDA-MB157 cells stably transfected with control or two individual Dub3 shRNAs and treated with or without 10 μM MG132 for 6 h was analysed by western blot. (**h**) Gastrulation and Snail1 expression were detected in *Drosophila* embryos and the mRNA was detected by real-time PCR using stage 11 cells (mean ± s.e.m. in three separate experiments).

cell renewal (up panel, Fig. 1h). This observation was very similar to that seen with a mutant Snail1 in *Drosophila* embryos, in which Snail1 is absolutely required for the dissociation and invagination of cells from epiblast[30]. Consistent with this observation, we noticed a drastic reduction of Snail1 in stage 11 cells. In addition, expression of several genes that are known to be repressed by Snail1 in this event, such as *Rho, Sim* and *T3*, were restored in embryos isolated from these RNAi lines (bottom panel, Fig. 1h). Together, these data indicated that Dub3 is specific for the control of Snail1 *in vivo*.

**Dub3 interacts with Snail1**. To further investigate the interaction of Dub3 with Snail1, we co-expressed Flag-Dub3 and HA-Snail1 in HEK293 cells and performed a co-IP experiment. After IP of Snail1, we detected an associated Dub3, and vice versa (Fig. 2a). IP of endogenous Snail1 and Dub3 from MDA-MB157 and MDA-MB231 cells also demonstrated the presence of endogenous Dub3 and Snail1, respectively (Fig. 2b). To identify the region in Snail1 that associates with Dub3, we generated two deletion mutants of Snail1 (refs 28,31): the N-terminal Snail1 (amino acids 1–153), which contains the SNAG domain of Snail1; and the C-terminal Snail1 (amino acids 153–264), which includes the conserved zinc finger motif (Fig. 2c). When these two deletion mutants of Snail1 were co-expressed with Dub3 in HEK293 cells, we found that the N-terminal region of Snail1 was responsible for its interaction with Dub3 (Fig. 2c). In addition, when GST-Dub3 was incubated with full-length or deletion mutants of Snail1, only the full-length and N-terminal domain of Snail1 were pulled down by GST-Dub3 (Fig. 2d).

Dub3 contains two functional domains; the N-terminal catalytic (UCH) domain and two hyaluronan binding motifs at its C terminus. To identify the region of Dub3 responsible for the interaction with Snail1, we generated a Myc-tagged full-length, N-terminal deletion, and C-terminal deletion of Dub3 (Fig. 2e) and co-expressed them with Snail1 in HEK293 cells. We found that the N-terminal catalytic domain retained the ability to interact with Snail1. However, when the C-terminal mutant was utilized, Dub3 was unable to interact with Snail1. When GST-Snail1 was pulled down, we found the presence of full-length and N-terminal Dub3 (Fig. 2f). Consistent with this, Dub3 only stabilized the N-terminal but not the C-terminal fragments of Snail1 (Fig. 2g). The interaction between Dub3 and Snail1 was further confirmed by immunofluorescence (IF) analysis showing that endogenous Dub3 co-localized with Snail1 in the nucleus of MDA-MB231 cells (Fig. 2h). Taken together, our results indicate that Dub3 interacts with Snail1 and that this interaction is mediated through the N-terminal regions of Dub3 and N-terminal region of Snail1.

**Dub3 stabilizes Snail1 through deubiquitination**. The interaction of Dub3 with Snail1 suggests that Dub3 regulates the protein stability of Snail1. To test this idea, we co-expressed Snail1 with Dub3 or vector control in HEK293 cells and examined Snail1 degradation. After treatment with cycloheximide to block newly protein synthesis, Snail1 degraded rapidly in cells transfected with a control vector (Fig. 3a). However, Snail1 levels were stabilized in the presence of Dub3 and this effect continued for up 4 h in the presence of cycloheximide. To test whether endogenous Snail1 is also subjected to similar regulation by Dub3, we knocked down endogenous Dub3 in MDA-MB231 cells, and found that endogenous Snail1 became unstable and degraded rapidly (Fig. 3b). To extend these findings and determine whether this Dub3 effect is mediated through a de-ubiquitination of Snail1, we co-expressed Flag-Snail1 with either WT- or CS-Dub3 in HEK293 cells. After

immunoprecipitating Snail1 from cells treated with MG132, we found that Snail1 was heavily ubiquitinated (lane 1, Fig. 3c). However, co-expression of WT-Dub3 almost completely abolished Snail1 ubiquitination while the CS-Dub3 did not have this effect (lanes 2 versus 3, Fig. 3c). Conversely, Snail1 ubiquitination significantly increased in Dub3-knockdown MDA-MB157 and MDA-MB231 cells after MG132 treatment (Fig. 3d). In an *in vitro* deubiquitination assay as described by Dupont et al.[32], we incubated poly-ubiquitinated Snail1 with purified WT-Dub3 or CS-Dub3. We found that WT-Dub3, but not CS-Dub3, specifically removed Snail1 ubiquitin moieties *in vitro* (Fig. 3e), indicating that Dub3 stabilizes Snail1 by removing its ubiquitination directly.

Previous studies showed that β-TRCP1 and FBXL14 are specific E3 ligases mediating the ubiquitination and degradation of Snail1 (refs 11,18,33). We investigated whether Dub3 stabilized Snail1 by impeding the activity of β-TRCP1 and FBXL14. Consistent with prior results, expression of β-TRCP1 and FBXL14 increased Snail1 protein degradation (lanes 4 and 7 versus lane 1, Fig. 3f). Expression of the WT-Dub3, but not CS-Dub3, blocked Snail1 degradation mediated by these two ligases. Conversely, knockdown β-TRCP1 or FBXL14 increased Snail1 stability (lanes 2 and 3, Fig. 3g). However, knockdown of Dub3 blocked the Snail1 stabilization effect mediated by the knockdown of either β-TRCP1 or FBXL14 (lanes 4 and 5, Fig. 3g), indicating that Dub3 is a critical factor controlling Snail1 stability. In agreement with this observation, expression of β-TRCP1 and FBXL14 increased Snail1 polyubiquitination (Fig. 3h), which was attenuated by expression of WT-Dub3 (lanes 3 versus 2, lanes 6 versus 5, Fig. 3h). Knockdown of β-TRCP1 or FBXL14 reduced Snail1 polyubiquitination, which was hampered by simultaneous knockdown Dub3 (Supplementary Fig. 2a). Both β-TRCP1 and FBXL14 share the same lysine pattern and target Snail1 degradation through ubiquitin modification of lysine 98, 137 and 146 (ref. 18). Consistent with previous reports, the Snail1 triple mutant (K3R) is more stable than WT-Snail1 (Supplementary Fig. 2b). However, ectopic expression of Dub3 still increased K3R accumulation, indicating that other lysines could be involved in Snail1 stability. Together, these data demonstrated that Dub3 counteracts β-TRCP1- and FBXL14-mediated Snail1 ubiquitination through deubiquitination.

**Dub3 expression induces EMT**. To study the functional effects of Dub3, we expressed Dub3 in two luminal breast tumour cell lines, MCF7 and T47D, which contain little endogenous Dub3 and Snail1 (Fig. 4a). Dub3 expression induced Snail1 stabilization as well as downregulation of E-cadherin and oestrogen receptor alpha (ERα) in these cells (Fig. 4a,b). Consistently, Dub3 expression induced a morphologic change indicative of EMT (Fig. 4b), including downregulation of epithelial markers (E-cadherin, Claudin-7 and Occludin) and the upregulation of mesenchymal molecules (N-cadherin and Vimentin) (Fig. 4c, Supplementary Fig. 3a). In addition, Dub3 expression converted these luminal cells into a basal-like phenotype; these cells lost luminal markers, such as ERα, FOXA1, CK18 and AGR2, and gained expression of basal molecules such as CK5, CD44 and EGFR (Fig. 4d, Supplementary Fig. 3a). We then tested the migration and invasiveness of these cells. Dub3 expression markedly increased the cell migration and invasive capacity (Fig. 4e,f, Supplementary Fig. 3b,c).

The catalytic activity of Dub3 is required for these functions, because CS-Dub3 could not induce Snail1 upregulation, or the morphological changes associated with EMT, or increased cell migration and invasion in these cells (Fig. 4a–f, Supplementary

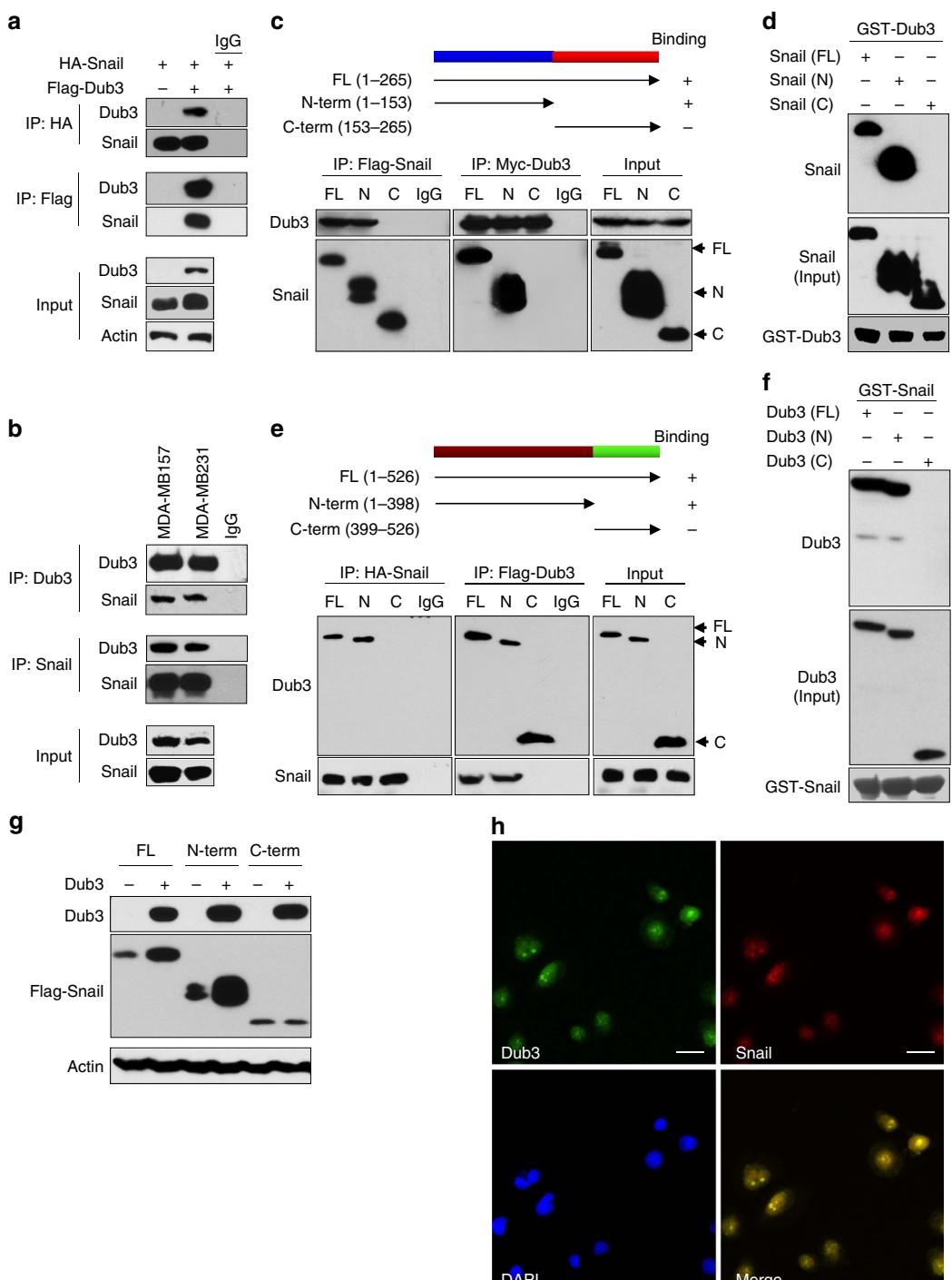

**Figure 2 | Dub3 interacts with Snail1.** (**a**) HA-Snail1 was co-expressed with vector or Flag-Dub3 in HEK293 cells. Snail1 and Dub3 were immunoprecipitated (IP) with HA or Flag antibody, respectively, and the associated Dub3 and Snail1 were analysed by western blot using either Flag or HA antibody. One-fortieth of the lysate from each sample was subjected to western blot to examine the expression of Snail1 and Dub3 (input lysate). (**b**) Endogenous Snail1 and Dub3 were captured by IP from MDA-MB231 and MDA-MB157 cells, and the bound endogenous Dub3 and Snail1 were examined by western blot. (**c**) Schematic diagram showing the structure of Snail1 and deletion constructs used (top panel). Flag-tagged full-length (FL) or deletion mutants of Snail1 were co-expressed with Myc-Dub3 in HEK293 cells. Extracts were subjected to IP with Flag or Myc antibody, and bound Dub3 or Snail1 was analysed by Western blot using either Myc or Flag antibody. (**d**) Lysates from HEK293 cells expressing WT or different deletion mutants of Flag-Snail1 were mixed with GST-Dub3. After pull-down by glutathione-agarose, the associated proteins were analysed by western blot. (**e**) Schematic diagram showing the structure of Dub3 and deletion constructs used (top panel). Myc-tagged full-length or deletion mutants of Dub3 were co-expressed HA-Snail1 in HEK293 cells. Extracts were subjected to IP with Myc or HA antibody, and the bound Snail1 or Dub3 was analysed by western blot using either HA or Myc antibody. (**f**) Lysates from HEK293 cells expressing WT or different deletion mutants of Myc-Dub3 were mixed with GST-Snail1. After pull-down by glutathione-agarose, the associated proteins were analysed by western blot. (**g**) Myc-Dub3 was co-expressed with Flag-tagged full-length or deletion mutants of Snail1 in HEK293 cells. The protein expressions of Dub3 and Snail1 were analysed by western blot. (**h**) Endogenous Dub3 and Snail1 in MDA-MB231 cells was detected by IF staining. Scale bars, 20 μm.

Fig. 3a–c). In addition, these functional activities promoted by Dub3 required Snail1 upregulation, because knockdown of Snail1 greatly inhibited these changes (Fig. 4a–f, Supplementary Fig. 3a–c). Together, these data indicate that Dub3 can induce EMT (luminal to basal-like phenotype conversion) by stabilizing Snail1 in breast cancer cells.

**Knockdown of Dub3 suppresses Snail1's function**. To further assess the function of Dub3 in breast cancer, we established stable clones with Dub3 knockdown in MDA-MB231 and MDA-MB157 cells. We achieved 80–90% knockdown efficiency

of endogenous Dub3 using two independent shRNAs (Fig. 5a). For both clones, Dub3-knockdown increased E-cadherin and Claudin-7 levels, downregulated expression of Vimentin and N-Cadherin, with concomitant changes of other EMT markers (Fig. 5a, Supplementary Fig. 4a). IF analysis also suggested a downregulation of E-cadherin and upregulation of Vimentin and N-cadherin (Fig. 5b). Dub3 knockdown greatly inhibited the migration and invasive capabilities of these cells (Fig. 5c,d, Supplementary Fig. 4b,c). Individual cell tracking also revealed Dub3 knockdown reduced the velocity and directionality of cell migration, and strongly inhibited the net distance of cell migration in MDA-MB231 and MDA-MB157 cells

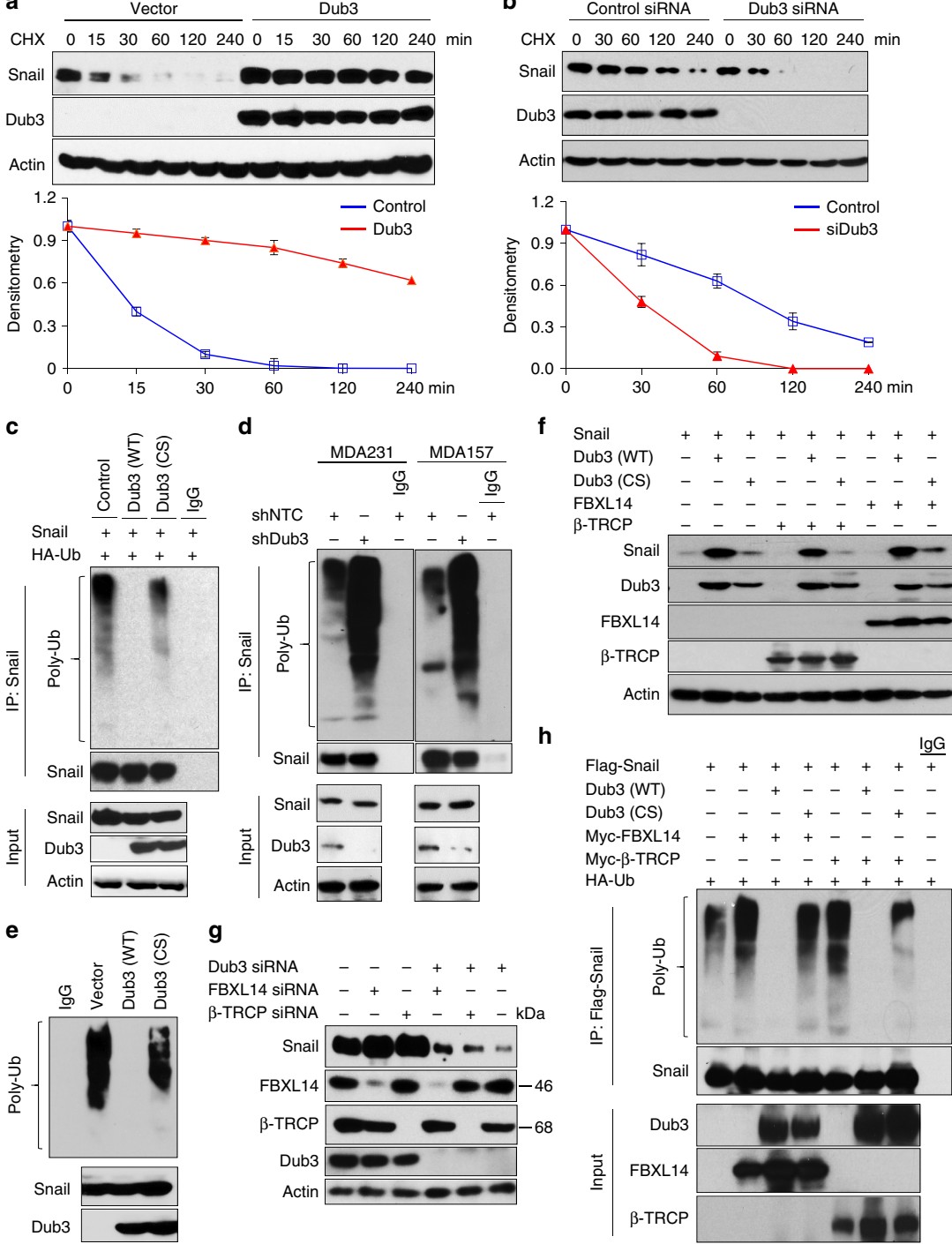

(Fig. 5e, Supplementary Fig. 4d). Importantly, Snail1-rescued expression partially inhibited E-cadherin and claudin-7 upregulation and increased Vimentin and N-cadherin expression in Dub3-knockdown MDA-MB231 and MDA-MB157 cells (Fig. 5a,b). Functionally, Snail1-rescued expression also restored migration and invasion in these Dub3-knockdown cell lines (Fig. 5c–e, Supplementary Fig. 4b–d).

MDA-MB231 and MDA-MB157 cells appear with stellate projections in 3D culture. Cells with Dub3 knockdown exhibited a marked change in morphology, with rounded/polygonal shape (Supplementary Fig. 4e). To extend assessment of the critical role of Dub3 in regulating CSC-like properties in human breast cancer, we examined tumorsphere formation in Dub3-knockdown clones. We found that Dub3 knockdown greatly reduced the number and size of primary and secondary tumorspheres in MDA-MB231 and MDA-MB157 cells (Fig. 5f, Supplementary Fig. 5a). This function of Dub3 is likely mediated through the regulation of Snail1, as Snail1 rescued expression (expressing Snail1-IRES-GFP) greatly restored the number and size of tumorspheres in these two cell lines. As human breast CSCs are enriched in a CD44$^{high}$/CD24$^{low}$ population[14,34–38], we measured this population in MDA-MB157 and MDA-MB231 cells with Dub3 knockdown using fluorescence-activated cell sorting (FACS). We found that Dub3 knockdown reduced the CD44$^{high}$/CD24$^{low}$ population in both cell lines (Fig. 5g, Supplementary Fig. 5b). To corroborate these findings, we also used a second set of breast CSC markers (CD49f$^{high}$/CD24$^{low}$)[39–42]. Similar to the results presented above, Dub3 knockdown reduced the population of CD49f$^{high}$/CD24$^{low}$ cells in MDA-MB231 and MDA-MB157 cells (lower panel in Fig. 5g, Supplementary Fig. 5c). Again, the reduction of a CSC population in Dub3-knockdown clones appears to be mediated by the downregulation of Snail1, as rescued Snail1 expression in Dub3-knockdown clones largely recovered the CSC phenotype. Taken together, these results clearly support our assessment that Dub3 is the crucial factor controlling Snail1 stability, EMT, migration and invasion, as well as CSC characteristics.

**Knockdown of Dub3 blocks breast cancer metastasis.** To directly assess whether Dub3 promotes metastasis *in vivo*, we intravenously injected Dub3-knockdown MDA-MB231 cells into female SCID mice and subjected these mice to bioluminescent imaging (BLI). Dub3-knockdown cells exhibited a reduced number of lung nodules at early time points (Fig. 6a,b), implying

that Dub3 is critical for the extravasation and/or colonization of breast tumour cells in lung. At 35 days post-injection, all control mice were moribund due to massive lung metastases with an average of 150 visible metastatic nodules per mouse (Fig. 6c,d). In contrast, mice injected with Dub3-knockdown cells were viable and free of detectable metastases. Histologic analyses supported the macroscopic observations and disclosed a high number of metastatic lesions produced by control cells whereas Dub3-knockdown cells lacked metastatic colonies (Fig. 6c,d). Consistent with the function of Snail1 *in vitro*, expression of exogenous Snail1 in Dub3-knockdown cells largely rescued the formation of lung metastases (Fig. 6a,d).

Snail1 is a key transcription factor of EMT[4,43]. To rule out the possibility of cellular adaptation effect associated with stable gene downregulation and to examine the temporal regulation of Snail1 *in vivo*, we generated a doxycycline (DOX)-inducible expression of Dub3 shRNA or control shRNA (TRIPZ lentiviral inducible shRNAmir system from Thermo Fisher Scientific) in MDA-MB231 cells. Treatment with DOX for 6 days achieved almost complete Dub3-knockdown and resulted in a remarkable downregulation of Snail1 (Fig. 6e). In an experimental metastasis model, we intravenously injected these cells into female SCID mice (left panel, Fig. 6f). Mice received DOX or no DOX in the drinking water 24 h after tumour cell inoculation. Dub3 knockdown after DOX treatment significantly decreased lung metastasis and lung weight, but these parameters showed no difference in control mice with or without DOX treatment (middle and right panels, Fig. 6f).

To further examine the therapeutic efficacy of systemic inhibition of Dub3 in preventing tumour recurrence and metastasis, we performed a spontaneous metastasis model analysis, in which control and DOX-inducible Dub3 shRNA MDA-MB231 cells were implanted into mammary fat pads of 6-week-old female SCID mice. When tumours reached a volume of 1 cm$^3$, the tumours was surgically removed. Mice then received DOX or no DOX in drinking water (left panel, Fig. 6g). Strikingly, the recurrent tumour was significantly inhibited in mice with the Dub3 shRNA expression (middle panel, Fig. 6g). In parallel, depletion of Dub3 also dampened spontaneous lung metastasis (right panel, Fig. 6g). Collectively, these data indicate that Dub3 facilitates breast cancer metastasis through, in large part, Snail1 stabilization.

**Dub3 is critical for IL-6-induced Snail1 stabilization.** We showed previously that IL-6 and TNFα can stabilize Snail1 by

**Figure 3 | Dub3 deubiquitinates Snail1 and antagonizes the function of Snail1's E3 ligase.** (**a**) Flag-Snail1 was co-expressed with vector or Myc-Dub3 in HEK293 cells. After treatment with cycloheximide (CHX) for the indicated time intervals, expression of Snail1 and Dub3 was analysed by western blot (top panel) using Flag and Myc antibodies, respectively. The intensity of Snail1 expression for each time point was quantified by densitometry and plotted (bottom panel). Experiment was repeated three times and a representative experiment is presented (mean ± s.e.m. in three separate experiments). (**b**) MDA-MB231 cells were transfected with control or Dub3 siRNA. After treatment with CHX as indicated above, expression of endogenous Snail1 and Dub3 was analysed by western blot (top panel); the intensity of Snail1 expression for each time point was quantified by densitometry and plotted (bottom panel) (mean ± s.e.m. in three separate experiments). Experiment was repeated three times and a representative experiment is presented. (**c**) Flag-Snail1 and HA-ubiquitin were co-expressed with WT or CS mutant Dub3 in HEK293 cells. After treatment with 10 μM MG132 for 6 hr, Snail1 was subjected to IP and the poly-ubiquitination of Snail1 assessed by western blot using HA antibody. IP Snail1 was blotted using Flag antibody. Input protein levels of Snail1 and Dub3 were examined using Flag and Myc antibodies, respectively. (**d**) MDA-MB231 and MDA-MB157 cells stably transfected with control, or Dub3 shRNA were treated with MG132 for 6 hr. Extracts were subjected to IP with Snail1 antibody and the poly-ubiquitination of Snail1 assessed by western blot using ubiquitin antibody. Input of Snail1 and Dub3 were analysed by western blot. (**e**) Ubiquitinated Snail1 was purified from MG132-treated HEK293 cells expressing Flag-Snail1, and then incubated with purified Myc-tagged WT-Dub3 or CS-Dub3 in a deubiquitination assay as described in Experimental Procedures. The poly-ubiquitinated state of Snail1 was assessed by western blot using HA antibody. The immuno-purified Snail1 and Dub3 used in this assay were analysed using Flag and Myc antibodies, respectively. (**f**) Flag-Snail1 was co-expressed with the indicated expression plasmids, and the expression of Snail1, Dub3, FBXL14, and β-TRCP1 were analysed by western blot. (**g**) MDA-MB231 cells were transfected with indicated siRNA and cell lysates were analysed by western blot. (**h**) Flag-Snail1 and HA-ubiquitin were co-expressed with indicated expression plasmids in HEK293 cells. After treatment with 10 μM MG132 for 6 h, Snail1 was obtained by IP and the poly-ubiquitination of Snail1 assessed detected by western blot using HA antibody. IP Snail1 was blotted using Flag antibody. Input protein levels for Dub3, FBXL14 and β-TRCP1 were assessed by western blot.

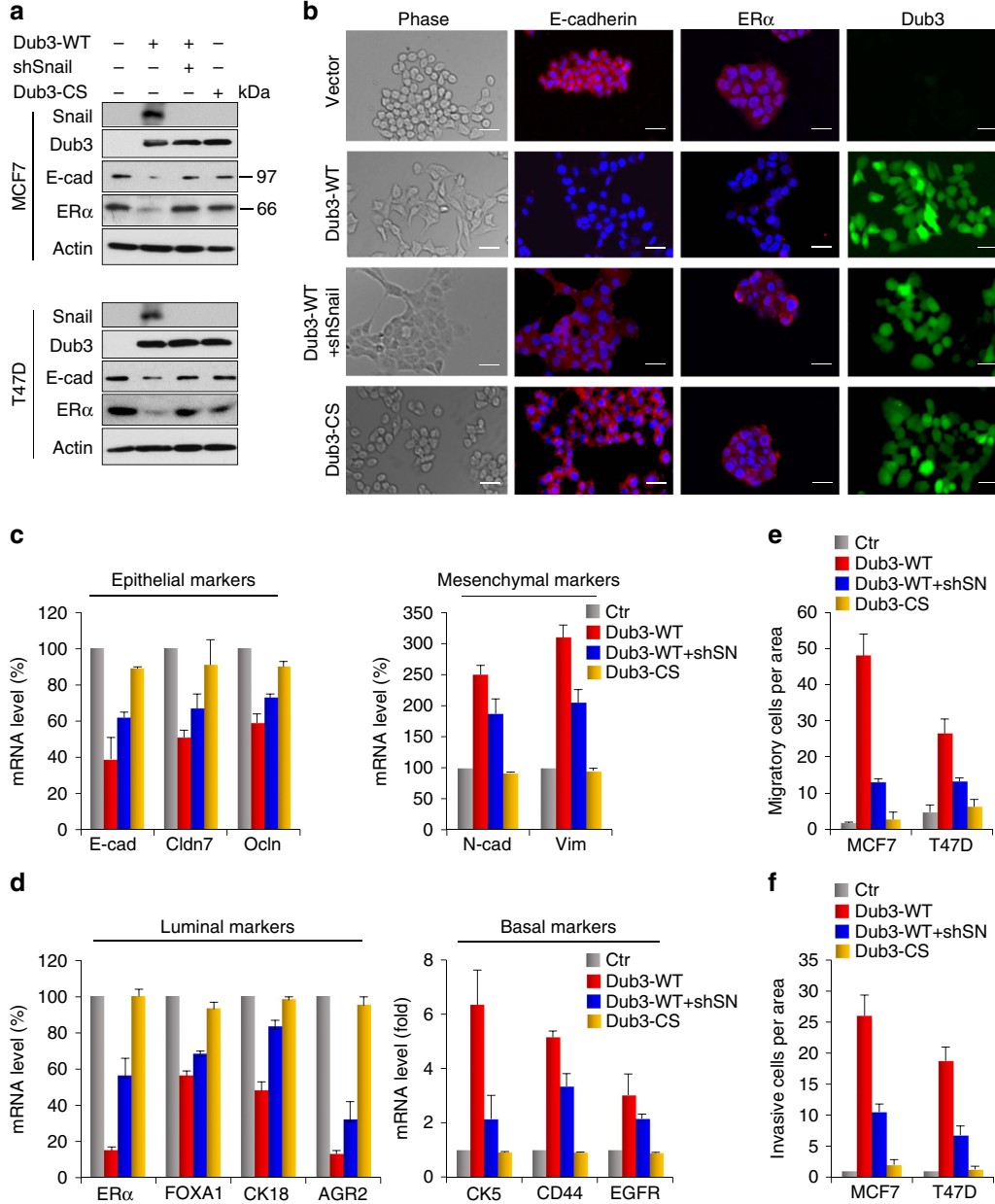

**Figure 4 | Overexpression of Dub3 induces EMT.** (**a**) WT- or CS-Dub3 was expressed in MCF7 and T47D cells. A rescue experiment with knockdown of Snail1 expression in WT-Dub3 expressing cells was also performed. The level of Snail1, Dub3, E-cadherin and ERα was analysed by western blot. (**b**) WT- or CS-Dub3 was expressed in MCF7 cells. A rescue experiment with knockdown of Snail1 expression in WT-Dub3 expressing cells was also performed. Morphologic changes indicative of EMT are shown in the phase contrast images; expression of E-cadherin, ERα and Dub3 was assessed by IF staining. Nuclei were visualized with DAPI (blue). Scale bars, 20 μm. (**c,d**) WT- or CS-Dub3 was expressed in MCF7 cells. A rescue experiment with knockdown of Snail1 expression in WT-Dub3 expressing cells was also performed. The mRNA levels of epithelial, mesenchymal (**c**), luminal, and basal (**d**) markers were quantitated by real-time PCR. Data are shown as mean ± s.d. of two separate experiments in triplicates. (**e**) Boyden chamber migration assay of modified MCF7 and T47D cells, as described in **a**. Data are presented as mean ± s.e.m. (**f**) Boyden chamber invasion assay of modified MCF7 and T47D cells, as described in **a**. Data are presented as mean ± s.e.m.

inhibiting the ubiquitination of Snail1, leading to EMT[19]. Interestingly, Dub3 was initially identified as an early response gene after stimulation by IL-6 and other cytokines[21,22]. These observations prompted us to investigate whether IL-6 induces Snail1 stabilization through Dub3 expression. We treated MDA-MB231 and MDA-MB157 cells with IL-6 (50 ng ml$^{-1}$) for different time intervals. Consistent with previous findings[22], Dub3 was rapidly induced in these two cell lines after 1 h of IL-6 stimulation (Fig. 7a). Snail1 was also robustly increased after

1 h of IL-6 stimulation and levels reached a maximum at 2 h. However, Snail1 mRNA levels showed no significant increase by 4 h of IL-6 treatment in these two cell lines (Supplementary Fig. 6a). In contrast, Dub3 knockdown in MDA-MB231 and MDA-MB157 cells not only reduced the endogenous level of Snail1 but also blocked IL-6-induced Snail1 stabilization (Fig. 7b).

The enzymatic activity of Dub3 is dependent on the ubiquitin carboxyl-terminal hydrolase (UCH) domain, which shares ∼50% sequence similarity (including strictly conserved catalytic

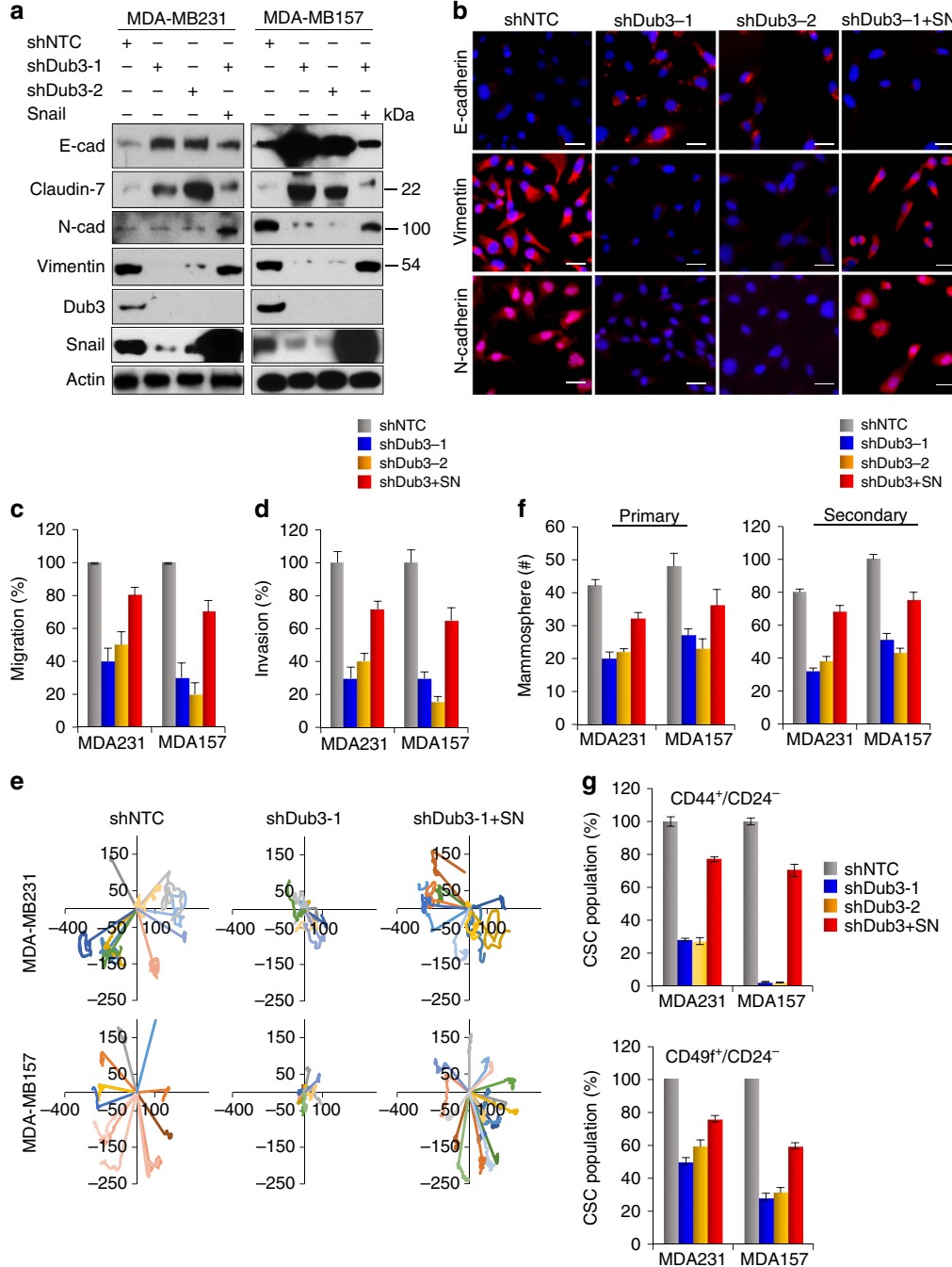

**Figure 5 | Knockdown of Dub3 inhibits migration, invasion and CSC-like characteristics in BLBC cells by downregulation of Snail1.** (**a**) Dub3 was knocked down by two different shRNA in MDA-MB231 and MDA-MB157 cells. Rescued Snail1 expression in these Dub3-knockdown clones were also performed. The expression of E-cadherin, Claudin-7, N-Cadherin, Vimentin, Dub3, and Snail1 was analysed by western blot. (**b**) IF images of EMT markers in MDA-MB231 cell lines described in (**a**). Scale bars, 20 μm. (**c**) Graphic representation of cell motility described in **a** analysed by a wound healing assay. Data are the percentage of migrating cells as the mean ± s.e.m. of three separate experiments. (**d**) Graphic representation of cell invasion described in **a**. Data are the percentage of vector control values (mean ± s.e.m. in three separate experiments in duplicates). (**e**) Cell trajectories of randomly selected cells described in **a**; each line indicates an individual cell's migration. (**f**) Graphic representation of primary and secondary tumorsphere-formation from cells described in **a** and are the mean ± s.d. from three independent experiments (left panel). (**g**) Graphic representation of the CD44high/CD24low (top) and CD49fhigh/CD24low population from cells described in **a** was examined by FACS analysis and are the mean ± s.e.m. from three independent experiments.

residues) with the UCH domain of USP2 (ref. 44), for which a structure has recently been reported (PDB access code 2HD5; please see 'Methods' for detail)[45]. We performed a docking analysis with several known DUB inhibitors and found that WP1130 could bind to the catalytic entry site of the UCH domain (left and middle panels, Fig. 7c)[46–48]. The physical interaction between recombinant Dub3 protein and WP1130 was further confirmed by an *in vitro* thermal shift binding assay[49]. As shown in Fig. 7c (right panel), WP1130 binding to Dub3 significantly shifted the melting temperature (Tm) of

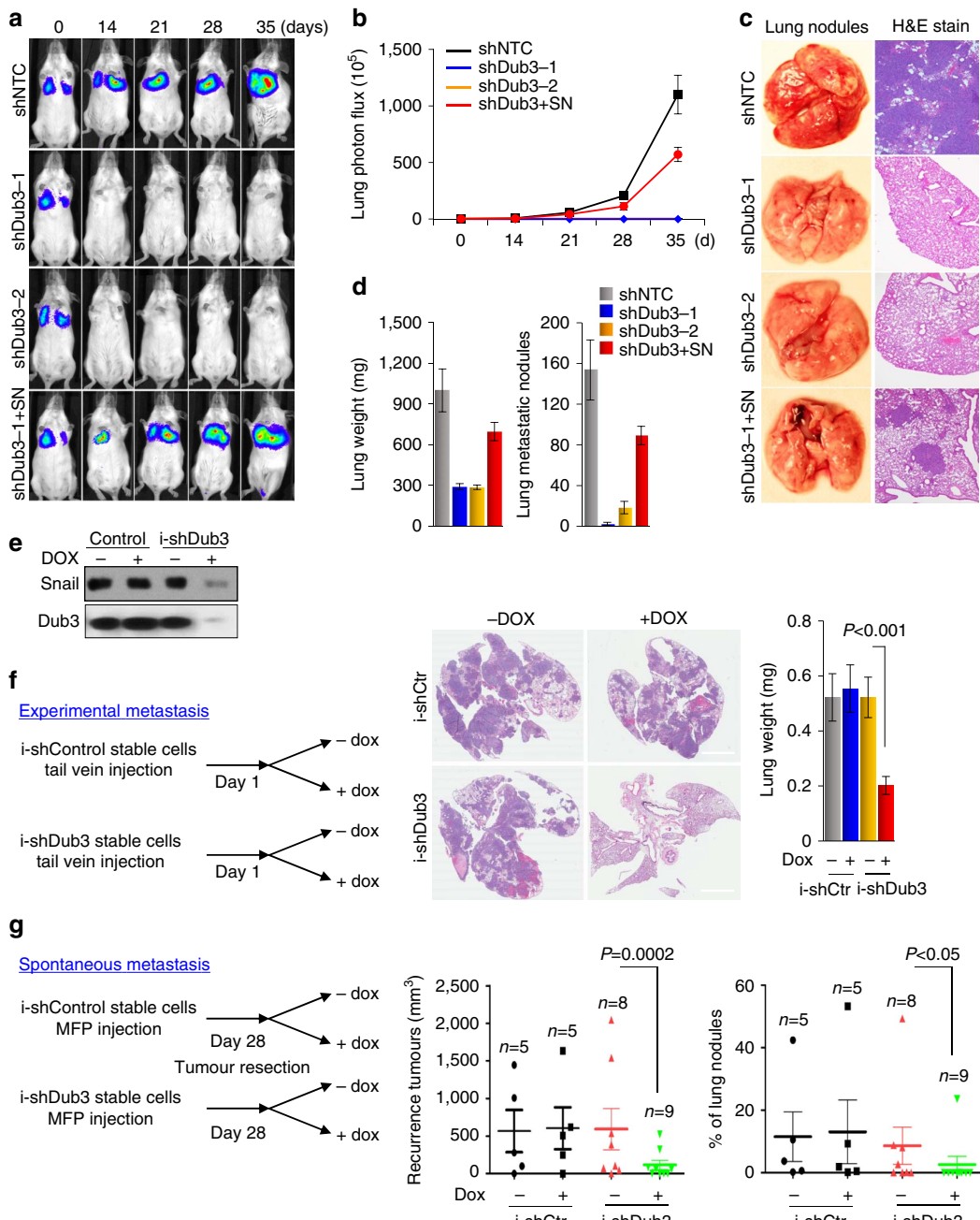

**Figure 6 | Knockdown of Dub3 inhibits tumour metastasis and recurrence of breast cancer *in vivo*. (a)** MDA-MB231-luc cells stably transfected with control, Dub3 shRNAs or Dub3-knockdown cells with Snail1 rescued expression were injected through tail vein into female SCID mice. Lung metastasis was assessed every week by bioluminescence imaging. Presented images are representative of each experimental group. (**b**) Normalized bioluminescence signals from lung metastasis in mice ($n = 6$) as experiment outlined above. Data are presented as mean ± s.e.m. (**c**) Representative images of lung lesions from experimental groups in **a**. (**d**) Graphic representation of lung weight and number of metastatic nodules from mice in experimental groups described in **a**. Data are presented as mean ± s.e.m. (**e**) MDA-MB231 cells stably transduced with Dub3-inducible shRNA were treated with or without DOX. Expression of Snail1 and Dub3 were analysed by western blot. (**f**) Schematic diagram outlining the experimental metastasis model (left panel). Images are the representative H&E stained lung sections (middle panel) and quantification of lung weight (right panel) from these mice. (**g**) Schematic diagram outlining the spontaneous metastasis model (left panel). Graphic representation of recurrent tumour size (middle panel) and metastatic lung nodules from these mice (right panel). For (**f,g**) *P* values were determined by Student's *t*-test. Data are presented as mean ± s.e.m.

Dub3 while the furan compound (negative control) had no effect under the same conditions. Negative Tm shifts (ΔTm) can be attributed to the compound destabilizing the protein or to the compound aggregating and causing early destabilization[50]. These types of negative shifts were observed for the compounds which contain heavy metal atoms, such as bromine (Br) in WP1130, and generate energetically unfavourable strains

when interacting with the proteins[51,52]. In addition, the direct binding between Dub3-UCH and WP1130 was demonstrated by the shifts during a native gel analysis in which similar dose-dependency and potential protein destabilization was observed (Supplementary Fig. 6b). These data clearly indicate that WP1130 physically interacts with Dub3 and can potentially alter its enzymatic activity. We thus

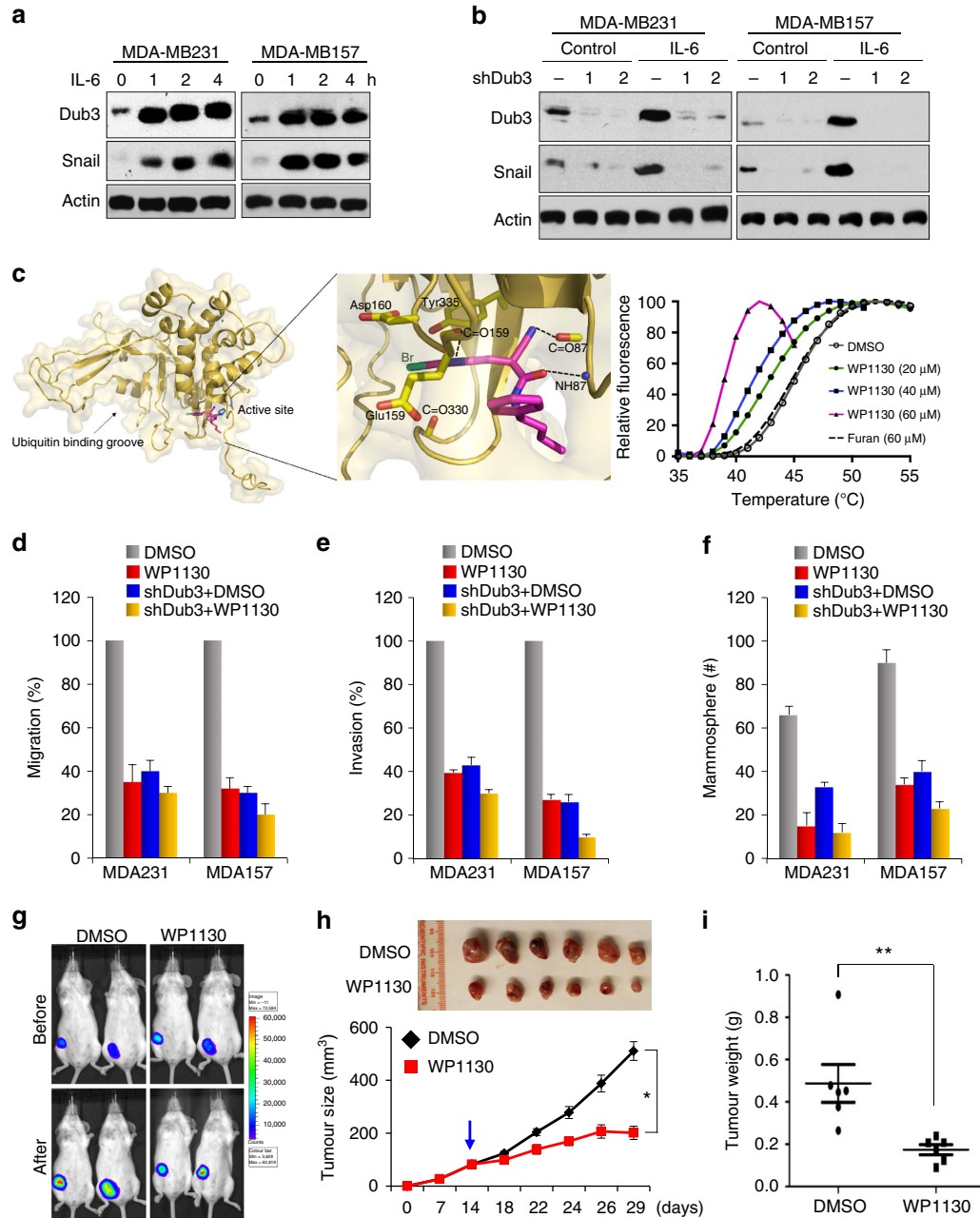

**Figure 7 | Dub3 is critical for IL-6 induced Snail1 stabilization.** (**a**) MDA-MB157 and MDA-MB231 cells were serum starved for 24 h, then treated with IL-6 for different time intervals. Expression of endogenous Snail1 and Dub3 were assessed by western blot. (**b**) MDA-MB231 and MDA-MB157 cells with Dub3-knockdown by two individual shRNAs or vector control were serum starved for 24 h followed by IL-6 treatment for 4 h. Expression of endogenous Snail1 and Dub3 were assessed by western blot. (**c**) A predicted model structure of Dub3 and WP1130 complex. The Dub3-UCH domain structure (yellow) is shown as ribbons while WP1130 (magenta) and the catalytic triad residues (sky blue: Cys89, His334, and Asp350) are shown as sticks. The inhibitor molecule binds to the opening surface of the active site where the ubiquitin cleavage site is presented. A zoom-up view of the interaction network is shown in the inset where the interacting residues are represented by sticks. These interactions include hydrogen bonds with backbone atoms (dotted lines) and potential halogen (Br) bonding interactions with the neighbouring electronegative atoms. Normalized melting curves (Right panel) depicting dose-dependent shifted thermal stability of Dub3 by WP1130 (coloured lines) from that of the apo protein (grey line), but not by a furan compound (dotted line; negative control). Tm for apo protein and $\Delta$Tm values for compound-bound proteins are indicated. (**d**) MDA-MB231 and MDA-MB157 cells stably expressing control vector or Dub3 shRNA were treated with 0.5 µM WP1130 for 24 h and analysed for cell migration using a wound healing assay. Graphic representation is the percentage of migration (mean ± s.e.m. in three separate experiments). (**e**) MDA-MB231 and MDA-MB157 cells stably expressing control vector or Dub3 shRNA were treated with 0.5 µM WP1130 for 4 h and analysed for cell invasion. Graphic representation is the percentage of invasive cells (mean ± s.e.m. in three separate experiments in duplicates). (**f**) MDA-MB231 and MDA-MB157 cells stably expressing control vector or Dub3 shRNA were treated with 0.5 µM WP1130 and analysed for tumorsphere-formation. Graphic representation is the number of tumorspheres (mean ± s.e.m. in three separate experiments). (**g,i**) MDA-MB231-luc cells were injected into the mammary fat pad of SCID mice. When tumours reached 100 mm³, mice were divided into two groups and treated with WP1130 (50 mg kg$^{-1}$) or solvent, respectively. Tumour size was recorded by bioluminescence imaging before or after 2-week of treatment (**g**). Tumour growth (**h**) and weights (**i**) were measured. Presented data are the mean ± s.d. from six mice. *P≤0.05, **P≤0.01; Student's t-test.

**Table 1 | Expression of Snail1 in different subtypes of breast tumour specimens.**

|  | Snail1 | | | | Total |
|---|---|---|---|---|---|
|  | Negative | Low | High |  |  |
| Non-TNBC |  |  |  |  | 169 |
| *Luminal (ER⁺) | 42 | 38 | 30 | 110 |  |
| †HER2⁺ | 15 | 26 | 18 | 59 |  |
| ‡TNBC (ER⁻, PR⁻, HER2⁻) | 21 | 33 | 111 |  | 165 |
| Total | 78 | 97 | 159 |  | 334 |

*$P = 0.210$, $R = -0.097$.
†$P = 0.210$, $R = 0.097$.
‡$P = 0.010$, $R = 0.488$.

**Table 2 | Expression of Dub3 in different subtypes of breast tumour specimens.**

|  | Dub3 | | | | Total |
|---|---|---|---|---|---|
|  | Negative | Low | High |  |  |
| Non-TNBC |  |  |  |  | 169 |
| *Luminal (ER⁺) | 36 | 45 | 29 | 110 |  |
| †HER2⁺ | 14 | 23 | 22 | 59 |  |
| ‡TNBC (ER⁻, PR⁻, HER2⁻) | 32 | 31 | 102 |  | 165 |
| Total | 82 | 99 | 153 |  | 334 |

*$P = 0.112$, $R = -0.123$.
†$P = 0.112$, $R = 0.123$.
‡$P = 0.010$, $R = 0.379$.

**Table 3 | Co-expression of Dub3 and Snail1 in different subtypes of breast cancer specimens.**

| Dub3 | Snail1 | | | | Total | |
|---|---|---|---|---|---|---|
|  | Negative | Low | High |  |  |  |
| Non-TNBC |  |  |  |  |  | 169 |
| *Luminal (ER⁺) |  |  |  |  | 110 |  |
| Negative | 17 | 12 | 7 | 36 |  |  |
| Low | 16 | 13 | 16 | 45 |  |  |
| High | 9 | 13 | 7 | 29 |  |  |
| †HER2⁺ |  |  |  |  | 59 |  |
| Negative | 3 | 8 | 3 | 14 |  |  |
| Low | 6 | 12 | 5 | 23 |  |  |
| High | 6 | 6 | 10 | 22 |  |  |
| ‡TNBC (ER⁻, PR⁻, HER2⁻) |  |  |  |  |  | 165 |
| Negative | 17 | 8 | 7 | 32 |  |  |
| Low | 2 | 15 | 14 | 31 |  |  |
| High | 2 | 10 | 90 | 102 |  |  |
| Total | 78 | 97 | 159 |  | 334 |  |

*$P = 0.265$, $R = 0.107$.
†$P = 0.424$, $R = 0.106$.
‡$P = 0.001$, $R = 0.643$.

treated MDA-MB231 cells with WP1130 and PR619, a non-selective inhibitor of the deubiquitinating enzymes[53]. Treatment of 0.5 µM WP1130 dramatically inhibited the intrinsic and IL-6-induced Snail1 stabilization while PR619 was less effective (Supplementary Fig. 6c). These results provide proof-of-concept that a Dub3 inhibitor will suppress the function of Snail1 by promoting its degradation; the findings also provide insight into an effective treatment modality for patients with BLBC. To further assess whether WP1130 treatment can inhibit Snail1 function, we first assessed the cytotoxicity of this compound in normal human breast epithelial (MCF10A) and in tumour (MDA-MB231) cell lines. Treatment with 1 µM WP1130 for up to 48 h, did not elicit any cytotoxicity in these cells (Supplementary Fig. 6d). We then treated the cells with 0.5 µM WP1130 and performed functional assays. We found that WP130 not only reduced tumour cell migration and invasion but also inhibited tumour mammosphere formation (Fig. 7d–f, Supplementary Fig. 7a–c). The suppressive effects of WP1130 are mainly mediated through Dub3 inhibition, because Dub3 knockdown greatly reduced the suppressive effects mediated by WP1130.

*In vivo* studies were performed by injecting MDA-MB231 cells into the mammary fat pads of NOD-SCID mice. When tumours were ~100 mm³, mice were divided into two groups to receive treatments of WP1130 or solvent control for two weeks. We found that WP1130 treatments significantly inhibited tumour growth (Fig. 7g–i). Taken together, these data indicate that the Dub3–Snail1 axis is the critical 'sensor-executor module' controlling EMT in response to microenvironmental signals.

**Dub3 and Snail1 are coordinately overexpressed in tumours.** To further examine the Dub3–Snail1 relationship in human breast cancer, we performed immunohistochemical (IHC)

analysis to examine Dub3 and Snail1 expression in a breast TMA generated by the Bio-specimen Repository in our Cancer Center at the University of Kentucky College of Medicine. The TMA contains 334 cases of breast tumour specimens, including 110 luminal, 59 HER2-overexpressing and 165 triple-negative breast cancer (TNBC) (Tables 1–3). Consistent with our observations in tumour cell lines, the intensity and distribution of Dub3 positively correlated with Snail1 in TNBC (Tables 1–3, Fig. 8a). We also found that Dub3 was upregulated in invasive tumour tissue compared with normal breast tissue from two gene expression datasets in Oncomine (Supplementary Fig. 8).

Because Snail1 expression predicts decreased relapse-free survival in women with breast cancer[54], we reasoned that women with primary breast cancers expressing high level of Dub3 relapsed at a faster rate than women whose breast cancers express low level of Dub3 in a pattern similar to that of Snail1. Therefore, we analysed two microarray expression datasets derived from primary human breast cancers in which both Dub3 expression level and clinical outcome were available. Intriguingly, individuals with high Dub3 expression had a significantly higher probability of developing distant metastasis and a reduced interval of disease-free survival (Fig. 8b). These results suggest that Dub3 expression may represent an important prognostic indicator for breast cancer in the clinical setting.

**Discussion**
Snail1 is a crucial transcription factor that plays an essential role in EMT, metastasis, CSC-like properties, metabolism and tumour recurrence. In this study, we found that Dub3 is a bona fide DUB for Snail1. The function of Dub3 is likely conserved from *Drosophila* to mammals, and knockdown of Dub3 increases, whereas Dub3 expression decreases, the ubiquitination and degradation of Snail1. The loss of Dub3 can be rescued by expressing exogenous Snail1. Most critically, a tight correlation between Dub3 and Snail1 on multiple cancer cell lines and human breast tumour specimens confirms their potential regulation. Our study provides several new insights into the involvement of ubiquitination in breast cancer metastasis. First, our study suggests that the Dub3–Snail1 signalling axis represents an important 'sensor-executor' module in breast cancer. It has

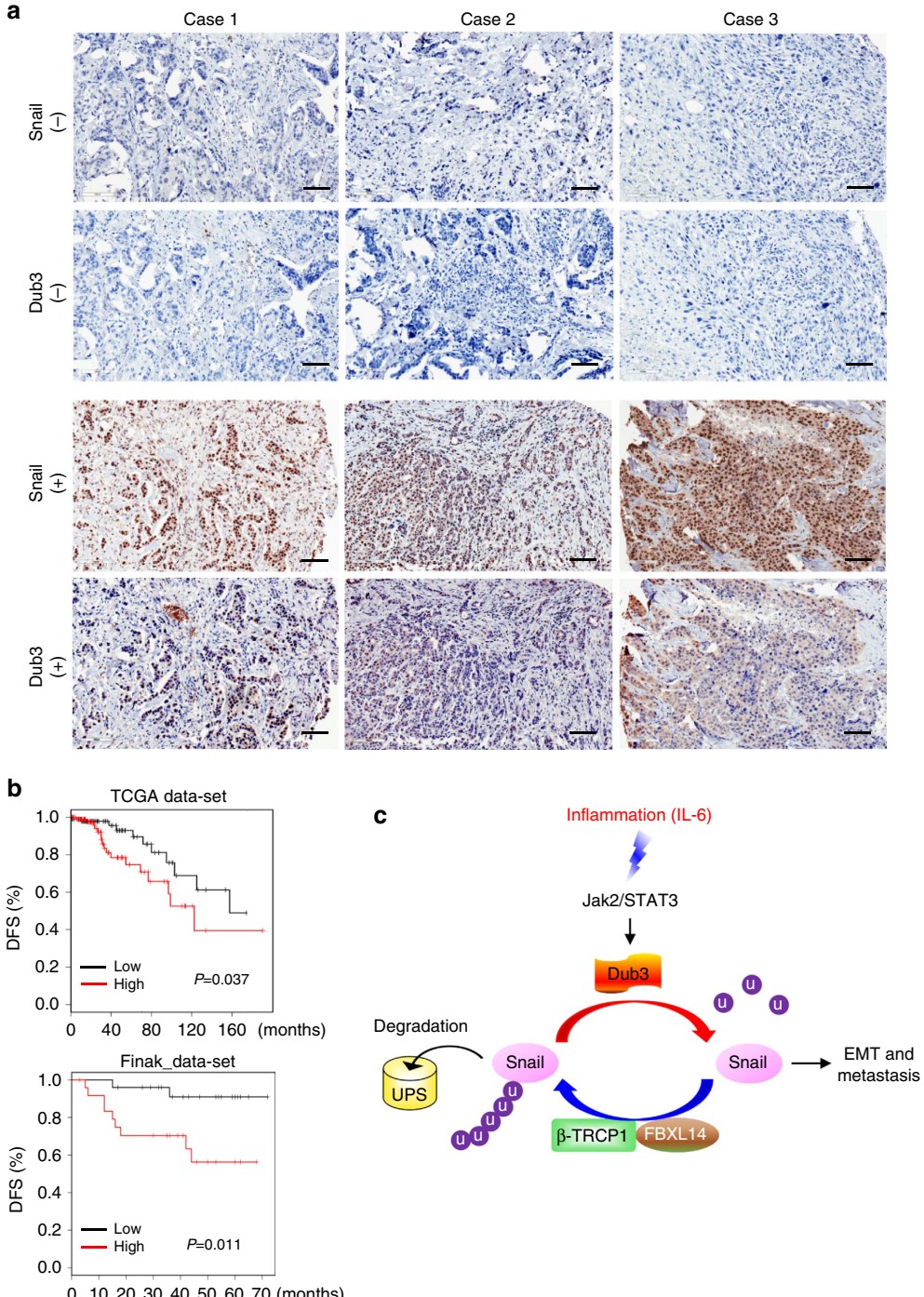

**Figure 8 | Expression of Dub3 and Snail1 are positively correlated in breast cancer patients.** (**a**) The 334 surgical specimens of breast cancer were immunostained using antibodies against Dub3, Snail1, and the control serum (data not shown). Images with consecutive IHC staining of both Dub3 and Snail1 in six cases of breast tumours (top panel: three cases of negative staining; bottom panel: three cases of positive) are shown (Scale bars, 50 μm). Statistical analysis is shown Tables 1–3. (**b**) Kaplan–Meier plots of distant metastasis-free survival of patients, stratified by expression of Dub3. Data obtained from the TCGA and Finak database. *P*-values represent log-rank testing of the difference in cumulative survival. (**c**) A proposed model to illustrate Dub3 induces Snail1 stabilization through a deubiquitination event. IL-6 induces the expression of Dub3, which antagonizes the function Snail1's ubiquitin E3 ligase (such as FBXL14 and β-TRCP1), leading to Snail1 stabilization and the acquisition of EMT and metastasis.

been noted that the migration and invasive capabilities of tumour cells at the invasive front are initiated and propelled by an inflammatory microenvironment through the induction of EMT. IL-6, a major cytokine present in the tumour microenvironment, can induce EMT and promote metastasis through the STAT3 signalling pathway in breast cancer, head and neck cancer and pancreatic cancer[55]. Elevated IL-6 level predicts tumour recurrence, poor response to chemotherapy, poor survival and tumour metastasis[56]. IL-6 is also identified as a major cytokine secreted by BLBC cells and is essential for the CSC-like characteristic of BLBC[57]. Therefore, it is likely that BLBC cells and infiltrated TAMs secrete IL-6 and provide

autocrine and paracrine feed-forward mechanisms, respectively, to sustain EMT and maintain CSC-like traits. Intriguingly, BLBC cells contain high levels of Dub3 and Snail1 and possess invasiveness and CSC-like characteristics. In contrary, the ERα-positive luminal subtype breast tumour cells (such as MCF7 and T47D), do not express IL-6 (ref. 55) and contain little Snail1. Interestingly, Dub3 is an early response gene of IL-6, and our study indicates that Dub3 is a critical deubiquitinase of Snail1. Therefore, Dub3 is one of the 'long-sought' missing molecule that senses extracellular inflammatory signals and converts them to Snail1 stabilization, which leads to the acquisition of CSC-like traits, invasion and therapeutic resistance in BLBC (Fig. 8c).

Second, our study indicates that Dub3 can block the activity of β-TRCP1 and FBXL14 to stabilize Snail1. Three E3 ligases have been identified that mediate Snail1 degradation. We previously demonstrated that GSK-3β phosphorylates Snail1 and promotes its nuclear export and interaction with β-TRCP1 (ref. 11). FBXL14, the human homologue of the Partner of Paired (Ppa) gene product which degrades Snail1 in *Xenopus laevis*, also degrades Snail1 in a phosphorylation-independent manner[18]. Recently, it has been shown that Snail1 can also be degraded by FBXO11 in a PDK1 phosphorylation-dependent manner in the nucleus[17,58]. It is likely that these different F-box containing E3 ligases function differently under diverse cellular contexts. We found that Dub3 can counteract the function of β-TRCP1 and FBXL14 by stabilizing Snail1. Intriguingly, both β-TRCP1 and FBXL14 can also modulate the degradation of other EMT-TFs, such Slug and Twist[18,59,60]. Whether Dub3 can also counteract the function of β-TRCP1 and FBXL14 in stabilizing Slug and Twist is a question that requires further investigation.

Third, our study indicates that Dub3 is an excellent therapeutic target for the inhibition of breast cancer metastasis and recurrence. Snail1 becomes stabilized and elevated in BLBC, but there is no clear ligand-binding domain for targeting Snail1, which creates a formidable obstacle for the development of small molecules to inhibit Snail1's functions. Our results indicate that Dub3 is a crucial molecule controlling inflammation-mediated Snail1 stabilization. Indeed, WP1130, which can bind to the catalytic entry site of the Dub3 UCH domain, blocked tumour cell migration, invasion and suppressed CSC-like properties. These data provide a proof-of-concept for therapeutic development of small molecules to inhibit the activity of Dub3 in metastatic breast cancer. Consistent with our findings, DUBs have emerged as a potential therapeutic target, given their role in several human diseases including cancer[61]. For example, the efficacy of a small molecule inhibitor of USP7 in multiple myeloma disease models provide the rationale for the development of next-generation USP-based therapies, and specifically demonstrates the promise of therapeutics targeting DUB to improve patient outcome[62]. Previously, Dub3 has been demonstrated to regulate both cell proliferation and G1/S cell-cycle progression and is increased in tumours. The current data strengthens the view that Dub3 is an ideal candidate for the development of potential inhibitors for cancer treatment based on the dual role of Dub3 in regulating cell growth and metastasis.

## Methods

**Plasmids and reagents.** Plasmids of wild-type and deletion mutants for Snail1 were generated as described[26]. The WT-Dub3 was from addgene. Dub3 (C89S) was generated using the QuikChange Mutagenesis kit (Stratagene, La Jolla, CA) as described previously[31]. All sequences were verified by DNA sequencing. Deletion mutants of Dub3 were constructed as described previously[31]. Antibodies used include: anti-Flag (F3165, 1:4,000, anti-Actin (A2228, 1:10,000), anti-Myc (9E10, 1:3,000) from Sigma-Aldrich (St. Louis, MO), Anti-Dub3 (Abcam, ab12991,

1:1,000); anti-Ub (Millipore, MAB1510, 1:500), N-cadherin (Upstate, 05-915, 1:1,000), anti-Snail1 (Cell Signaling, 4719, 1:1,000), Vimentin (Ab-2, 1:2,000) and ERα (Ab-15, 1:1,000) from Neomarkers, anti-HA (Roche, 3F10, 1:10,000), and anti-E-cadherin (610181, 1:10,000, BD Bioscience) and Claudin-7 (Abcam, ab27487, 1:1,000). Dub3 shRNA expression plasmids were purchased from MISSION shRNA at Sigma-Aldrich (St. Louis, MO). WP1130 and PR619 were from Selleck. Smartpool siRNA against human Dub3 was from Dharmacon (Chicago, IL).

**Cell culture.** The human embryonic kidney HEK293, breast cancer MDA-MB231, MDA-MB157, MCF7, SKBR3, and colon cancer HCT116, HT-29 cell lines were purchased from the American Type Culture Collection (Manassas, VA) and grown in Dulbecco's modified Eagle's/F12 medium plus 10% fetal bovine serum as described previously[26]. Breast cancer cell lines (T47D, ZR75, BT474) and prostate cancer cell lines (LNCaP, Du145, PC3) were grown in RPMI1640 plus 10% FBS. The culture medium for SUM159 and SUM149 is Ham's F-12 (Invitrogen) supplemented with 5% FBS, 5 µg ml⁻¹ insulin, and 1 µg ml⁻¹ hydrocortisone (Sigma, St. Louis, MO). All the cells lines are routinely checked for morphological and growth changes to probe for cross-contaminated, or genetically drifted cells. If any of these features occur, we use the short tandem repeat (STR) profiling service by ATCC to re-authenticate the cell lines.

**Small interfering RNA library screening.** The human deubiquitinating enzyme siGENOME RTF Library was purchased from Dharmacon (Chicago, IL). The screen was performed according to manufacturer's instructions. In brief, the cells were added to the rehydrated Dharmacon RTF siRNA library plates. Two days later, the cell lysates were extracted and the expression of Snail1 was detected with western blot.

**Invasion assay.** Invasion assays were performed in Boyden chambers coated with Matrigel as instructed by the manufacturer (BD Biosciences, San Jose, CA). Various cancer cell lines were seeded on the top of the Matrigel in the upper chamber while the bottom chambers were filled with non-serum culture medium plus 100 nM LPA. The invasive cancer cells were stained with crystal violent. All experiments were performed in triplicate.

**Single-cell migration assay.** Cells were seeded on glass-bottomed dishes (MatTek, Ashland, MA, USA) that had been coated with 5 µg ml⁻¹ fibronectin. Real-time images were taken under Nikon Biostation IMQ Cell every 10 min for 6 h. The movement of individual cells was analysed using NIS-Element AR Software (Nikon), and the distance that was travelled during time was measured as indicated.

**GST pull-down assay.** Glutathione-S-transferase proteins were expressed as described previously[31]. Cells were subjected to lysis in GST pull-down buffer (20 mM Tris, 150 mM NaCl and 1% Nonidet P-40 with protease cocktail) and rotated with glutathione–Sepharose-bound GST-Snail1 or GST-Dub3. The binding complexes were eluted with SDS–PAGE sample buffer. About one-tenth of these eluents were analysed by western blot and the rest were examined for the presence of purified GST protein by Coomassie Blue staining.

**Immunoprecipitation and western blotting.** For protein extraction, $5 \times 10^5$ cells per well were plated onto six-well plates and transiently transfected with indicated expression plasmids. At 48 h after transfection, cells were incubated with or without the proteasome inhibitor MG132 (10 µM) for an additional 6 h before protein extraction and western blot analysis. Primary antibodies against Flag (M2, 1:1,000) and HA (3F10, 1:4,000) were used for protein detection. For IP, HEK293 cells transfected with the indicated expression plasmids were lysed in buffer (50 mM Tris (pH 7.5), 150 mM NaCl, 5 µg ml⁻¹ aprotinin, 1 µg ml⁻¹ pepstatin, 1% Nonidet P-40, 1 mM EDTA and 0.25% deoxycholate). Total cell lysates (1,000 µl) were incubated overnight with 1 µg of anti-HA or anti-Flag antibody conjugated to agarose beads (Roche Molecular Biochemicals) at 4 °C. The beads were then washed with lysis buffer, and the immunoprecipitated protein complexes were resolved by 10% SDS–PAGE. Some important original immunoblotting results are shown in Supplementary Fig. 9.

**Immunofluorescence staining.** For IF microscopy, cells were grown on cover slips, fixed with 4% paraformaldehyde and incubated overnight with anti-Dub3 and anti-Snail1 antibodies. Proteins were visualized by incubation with goat anti-mouse conjugated with Alexa fluor 568 and goat-anti-rabbit conjugated with Alexa fluor 488, respectively (Invitrogen, Carlsbad, CA). Finally, cover slips were incubated with 4′,6′-diamidino-2-phenylindole (Sigma-Aldrich) for 20 min and visualized under a fluorescent microscope.

**Immunohistochemical staining.** Breast cancer tissue microarray (TMA) of 334 cases of invasive ductal carcinomas is obtained from the tissue bank at the Markey Cancer Center's tissue repository at our institute. Tissue samples were

stained with anti-Dub3 (Abcam, ab12991, 1:100 dilution) and anti-Snail1 (Abcam, ab53519, 1:250 dilution) antibodies, and each sample was scored by an *H*-score method that combines the values of immunoreaction intensity and the percentage of tumour cell staining as described previously[19]. Chi-square analysis was used to analyse the relationship between Dub3 and Snail1 expression; statistical significance was defined as $P < 0.05$.

**Quantitative real-time PCR.** Total RNA was isolated using RNeasy Mini kit (Qiagen) according to the manufacturer's instructions. Specific quantitative real-time PCR experiments were performed using SYBR Green Power Master Mix following manufacturer's protocol (Applied Biosystems).

**Fluorescence-activated cell sorting.** Cells were detached from plates and incubated with anti-human CD44 and anti-human CD24 (PE-conjugated, ebioscience) or anti-human CD49f (PE/Cy7 CD49f, e-Bioscience), finally analysed using a FACSCalibur flow cytometer.

**Tumorsphere formation assay.** Tumorsphere cultures were performed as described in Dontu et al.[63]. In brief, Cell monolayers were plated as single-cell suspensions on ultra-low attachment plates (Corning) in DMEM/F12 medium supplemented with 20 ng ml$^{-1}$ EGF, 10 μg ml$^{-1}$ insulin, 0.5 μg ml$^{-1}$ hydrocortisone and B27. Tumorspheres were counted via visual inspection after 5–10 days.

***In vivo* ubiquitination assay.** HEK293 cells were transfected with HA-ubiquitin, Flag-Snail1 and Myc-Dub3 plasmids as indicated. The cells were treated for 6 h with 10 μM MG132 at 48 h post transfection, and then lysed. The samples were immunoprecipitated using anti-Flag agarose (Sigma).

***In vitro* deubiquitination assay.** The *in vitro* deubiquitination was performed as described[32]. Briefly, HA-ubiquitin and Flag-Snail1 were co-expressed in HEK293 cells. After cells were treated with 10 μM MG132 for 6 h, ubiquitinated Snail1 was isolated by IP with Flag antibody. In a parallel experiment, Myc-Dub3 (WT or CS) or vector was expressed in HEK293 cells, and purified by IP with anti-Myc Affinity Matrix (Roche, USA). The purified Dub3 was eluded with Myc peptide, dialyzed and subsequently incubated with ubiquitinated Snail1 in a deubiquitination reaction buffer (50 mM HEPES, pH 7.5, 100 mM NaCl, 5% glycerol, 5 mM MgCl₂, 1 mM ATP and 1 mM DTT) at 30 °C. The ubiquitinated status of Snail1 was analysed by western blot with HA antibody.

**Complex model structure of Dub3 and WP1130.** For protein–ligand docking studies, the three-dimensional (3D) structure of Dub3-UHC was built by comparative protein structure modelling from the homologous USP2 crystal structure (PDB access code 2HD5) as a template using the programme MODELLER[64]. The WP1130 atomic coordinates were generated using the stereochemistry information stored in PubChem. The complex structure was modelled using the SwissDock protein-small molecule docking simulation software[65]. This software adopts the CHARMM simulation programme[66], which preforms numerous conformational and path sampling methods, free energy estimates, molecular minimization, dynamics and analysis techniques. This programme has been known to be highly successful for small and relatively rigid ligands with < 10 flexible rotatable bonds.

**Fluorescence based thermal shift assay.** Purified recombinant Dub3 protein was used to screen small molecule compounds in a fluorescence based thermal shift assay[49]. Dub3 protein was dispersed in a buffer containing 20 mM HEPES, pH 7.0 and 150 mM NaCl. The final protein concentration in a 20 μl reaction volume was 10 μM. Ligands to be tested were added at 2 ×, 4 ×, or 6 × concentration such that the DMSO concentration never exceeded 2%. SYPRO Orange dye (Invitrogen) was added last at a 5 × concentration. The PCR tubes were then sealed, centrifuged and heated from 25 to 95° at a rate of 1° per min on 7500 Real-Time PCR machine (Applied Biosystems). Raw data analysis and curve fitting to calculate Tm values was performed as described.

***In vivo* tumorigenesis assay.** Female SCID mice (6–8 week old) were purchased from Taconic (Germantown, NY) and maintained and treated under specific pathogen-free conditions. All procedures were approved by the Institutional Animal Care and Use Committee at the University of Kentucky College of Medicine and conform to the legal mandates and federal guidelines for the care and maintenance of laboratory animals. Mice were injected with the breast cancer MDA-MB231-luc cells and corresponding stable clones with knockdown of Dub3 or Snail1 expression ($5 \times 10^5$ cells per mouse, 6 mice per group) via tail vein injection. Lung metastasis was monitored by the IVIS bioluminescence imaging system.

For the spontaneous metastatic model, mice were injected with the breast cancer MDA-MB231-luc cells and corresponding inducible stable cells via mammary gland fat pad. The growth of the primary tumour was monitored by external caliper measurement once a week. When tumours were ∼1 cm³, the

primary tumour was surgically removed and the incision was closed with wound clips. The mice were randomly separated into two groups and treated with or without doxycycline in the drinking water. Animals were euthanized 5 weeks after primary tumour removal to investigate the development of pulmonary metastasis.

For animals subjected to drug treatment, MDA-MB231-luc cells were injected into the mammary gland fat pad of 8-week-old female SCID mice. Tumour growth was monitored with caliper measurements. When tumours were ∼100 mm³ in size, WP1130 was administered every other day for 2 weeks. Data were analysed using the Student's *t*-test; a *P* value < 0.05 was considered significant.

**Patient samples.** The frozen fresh tumour samples were collected from resected breast tumours from patients at our institute with the approval of the Institutional Review Board. These frozen samples were 'snap-frozen' in liquid nitrogen and stored at − 80 °C. Each sample was examined histologically with hematoxylin and eosin (H&E) stained sections. Regions from tumour samples were microdissected and examined. Only samples with a consistent tumour cell content of more than 75% in tissues were used for analysis. Samples were then homogenized using 20 strokes of a Dounce homogenizer in 1 ml of homogenizing buffer. Following centrifugation, pellets were re-suspended in lysis buffer and processed for western blot.

**Data availability.** The data that support the findings of this study are available from the corresponding authors upon reasonable request.

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

## Acknowledgements

We thank Dr Cathy Anthony for critical reading and editing of this manuscript. We also thank Drs Diaz VM and de Herreros AG for providing Snail1 K3R and FBXL14 plasmids. This research was supported by the Shared Resources of the University of Kentucky Markey Cancer Center (P30CA177558 to B.M.E.). This work was also supported by grants from NIH (CA125454 and CA188118), DoD (BC140733P1), Mary Kay Ash Foundation and the Frankfort Country Club's Ladies Golf Association (to B.P. Zhou), and American Cancer Society Research Scholar Award (RSG13187) (to Y.Wu).

## Author contributions

Y.Wu and B.P.Z. conceived and designed the study. Y.Wu, Y.Wang, Y.Lin, Y.Wang and C.D. performed most of the study. Y.Liu and J.J. performed the study on *Drosophila* analyses. D.N. and Q.S. help on the IHC and tumour sample analyses. P.S. and Y.-I.C. performed Dub3 protein purification and binding analyses. C.W. performed bio-informatic and statistical analyses. J.D., W.L. and M.T. and B.M.E. discussed the results, conceived some experiments, and provided critical reagents and comments. Y.Wu and B.P.Z. wrote the manuscript.

## Additional information

**How to cite this article**: Wu, Y. *et al.* Dub3 inhibition suppresses breast cancer invasion and metastasis by promoting Snail1 degradation. *Nat. Commun.* **8**, 14228 doi: 10.1038/ncomms14228 (2017).

