## [Peer Review File · Nature Communications]

Reviewers' comments:

Reviewer #1 (Remarks to the Author): expert in ubiquitination and cancer

Snail is a key factor controlling EMT and cancer metastasis. While E3 ligases responsible for Snail ubiquitination and degradation have been defined, the deubiquitinating enzyme (DUB) removing Snail ubiquitination remains not well understood. Zhou and his colleagues in this study provide convincing evidence that Dub3 is a bona fide DUB for Snail and stabilizes Snail by removing Snail ubiquitination. Notably, Dub3 was shown to promote EMT and breast cancer metastasis through promoting Snail stability. Overall, the study is novel and the conclusion is largely supported by experimental data. The study reveals not only new insight into how Snail deubiquitination process is orchestrated, but also suggests that Dub3 is a potential target for the treatment of breast cancer metastasis. However, there are some issues remain to be addressed.

1. The authors provide promising evidence in Figure 4 that Dub3 regulates EMT, migration and invasion. However, it is unclear whether such effects are dependent on its deubiquitinating activity.
2. DUB3 inhibition is shown to inhibit breast cancer metastasis. However, it is unclear whether DUB3 overexpression promotes breast cancer metastasis. If so, is its DUB activity required for this process?
3. In Fig 3F, the authors claimed that beta-TRCP promoted Snail degradation. However, the reviewer found no such effect by beta-TRCP (lane 1 & Lane 7) . The authors need to repeat the experiment in order to draw the precise conclusion.
4. Using computer modeling and in vitro binding assay, DUB inhibitor WP1130 was found to bind to Dub3. Subsequent experiments revealed that WP1130 inhibited IL-6-mediated Snail stability, cell migration, tumor sphere formation and primary tumor formation. Based on these results, the authors concluded that WP1130 displayed such effect likely through Dub3 inhibition. However, there is no direct evidence that WP1130 inhibits Dub3 activity. Since WP1130 also inhibits other DUBs, it is unclear whether WP1140 indeed acts through Dub3 inhibition to impact on those phenotypes. To make such claims, the authors may examine these phenotypes in the context of Dub3 deficiency.

Reviewer #2 (Remarks to the Author): expert in breast cancer, EMT and metastasis

In this manuscript, Wu and coauthors report that Dub3 is a protein deubiquitinase that is involved in Snail1 stabilization. The authors are able to identify Dub3 specific binding to Snail1, mapping the domains in both proteins responsible for this interaction. Moreover, they find that Dub3 and Snail1 expression correlate in several cancer cell lines while Dub3 stabilizes Snail1 particularly in basal breast cancer cell lines where both proteins are expressed and co-localize. Mechanistically they propose that in this particular scenario, Dub3 stabilizes Snail1 post-transcriptionally deubiquitinating Snail1 and counteracting the action of known E3 ligases responsible for Snail1 proteasome-mediated degradation. Moreover, Dub3 ectopic expression in luminal breast cancer cell lines promotes EMT and associated features that the authors claim are mediated through Snail1 upregulation. Additionally, the authors show that Dub3 depletion blocks lung colonization by breast cancer cell lines in vivo, both in experimental and spontaneous metastasis assays. Finally, they claim that IL6 is responsible for increasing Dub3 levels and subsequent Snail stability whereas a Dub3 inhibitor is able to reduce primary tumor growth of breast cancer xenografted cells in immunocompromised mice. Wu et al propose Dub3 as a poor prognostic factor in human breast cancer.

MAJOR POINTS

The manuscript is very well written, results and figures have remarkable quality and most results are backed by sound experiments, whereas the major conclusions drawn by the authors entail sufficient novelty and are sustained by the data shown herein.

I would recommend accepting the manuscript once the authors revise some specific concerns and few minor points.

- 1) Regarding the Abstract I'd suggest revising the sentence "knockdown of Dub3 resulted in Snail1 destabilization, suppressed EMT, and decreased tumor initiation and metastasis. These effects were rescued by ectopic Snail1 expression". This is not accurate in regards to the results shown herein since Dub3 knockdown might revert EMT and decreases metastasis whereas Dub3 inhibition by WP1130 decreases tumor initiation. Snail1 expression rescues lung colonization but not primary tumor growth.
- 2) The data involving *Drosophila* development while interesting might dilute the message. Do the authors consider Dub3 is the unique DUB stabilizing Snail1, regardless of the context?
- 3) In Figure 4 the IF images should be improved. Same cell confluence should be shown to properly address the EMT phenotype. Moreover, better image quality regarding E-cadherin subcellular localization is necessary since that shown does not localize properly, at least in MCF7 cells. The same accounts for Fig. 5b. E-cadherin localization is not adequate in shDub3 cells since it even seems nuclear. Besides, to properly conclude in line 200 "Together, these data indicate that Dub3 can induce EMT (luminal to basal like phenotype conversion) by stabilizing Snail1 in breast cancer cells" the authors would need to silence Snail in cells MCF7 and T47D that overexpress Dub3. This way they would be able to conclude that all the effects shown upon Dub3 expression are indeed mediated exclusively by Snail1.
- 4) I'd strongly recommend not using shDub3 selected clones and repeat the experiments shown in Figure 5 with pools of cells recently infected and silenced for Dub3. Moreover, the levels of Snail in MD231 shNTC cells in Fig. 5a are extremely low compared to other panels.
- 5) Regarding stemness it is surprising to find such a huge proportion of CSC within breast cancer basal cells (around 70%). I'd suggest further reviewing the literature cited and discuss their results since probably this population is fibroblastic cells and not CSC. To state that, further experiments are required, such as limiting dilution tumorigenicity assays, etc. I am also little convinced about the mammosphere forming ability shown in Fig. 5. In order to be able to conclude that, serial passaging of mammospheres should be performed to assess self-renewal ability.
- 6) In regards to WP1130 results, it is necessary to present data concerning the cytotoxicity assays it is published that that WIP1130 has pro-apoptotic activity and those data are necessary if the authors suggest its use as possible cancer treatment.
- 7) The results regarding Snail1 IHC are intriguing. Snail1 IHC in tumor samples has remained elusive for many years and in many laboratories so it'd be convincing to show several images of different samples showing Snail1 by IHC. Besides, the authors are vague about these tumor samples and they should clarify which kind of breast tumor samples are these. Coexpression would be expected in TNBC but not in all breast tumor samples according to the authors' hypothesis. It might be worth to show Snail and Dub3 association to different types of breast tumor types. Moreover, Fig. S8 is not clear at all. And its Legend refers to expression of both Snail and Dub3 which is not mentioned within the text.

MINOR POINTS

1. Line 88: it doesn't seem adequate to state "To understand the regulation of Snail1 in breast cancer, we purified the Snail1 complexes from nuclear extracts of 20 liters HeLa" since HeLa are not breast cancer cells. I'd suggest rephrasing.
2. Figure 2c: the authors should mention why there are two clear bands when N-terminal Snail1 is cloned; whereas the results regarding binding to the C-terminal part of Dub3 are not conclusive due to the low expression of this construct.
3. Line 138: Do the authors really did this: "We also purified full-length Dub3 from a GST-Dub3 fusion protein by cleaving and removing the GST portion with the tobacco etch virus (TEV) protease".
4. Line 173: the references 32 and 33 do not seem adequate herein.
5. Line 207: the sentence is not adequate, Dub3 knockdown doesn't restore E-cadherin levels, it increases them.
6. Line 287: S7b instead of S7c.

Reviewer #3 (Remarks to the Author): expert in cancer and molecular modelling

Reviewer's Comments. Dub3 Inhibition Suppresses Breast Cancer Invasion and Metastasis by Promoting Snail1 Degradation, Wu et al.

Description. The purpose of this study was to demonstrate that Snail1 protein, that is critical in inducing epithelial to mesenchymal cell conversion, is stabilized by a de-ubiquitinating protein, Dub3, that directly binds to it. In the absence of this protein, Snail1 is subjected to ubiquitination by such proteins as FBXL14 and beta-TRCP and targeted for proteolysis in the proteasome. Dub3 protein induces cell migration and metastasis in several human tumors making it a desirable target for anti-cancer agents.

Critique.

1. The initial experiments, the results of which are described in Figures 1 and 2, are poorly defined and give results that even contradict the contention of the authors and the results presented further on. The protocols for Figures 1 are undefined. Are the authors transfecting constructs for FLAG-Snail1 and HA-Dub3 into HEK 239 cells and then immunoprecipitating each? If, as the authors document further on, CS-Dub3 is inactive, why is there a rather high level of Snail1 in Fig 1A? Also, how is the intracellular concentration of Dub 3 increased in Fig 1B? Is its expression under the control of an inducer? If so what is the construct? This has to be defined.

2. Fig1 e is supposed to show that in a variety of human tumors Snail1 expression correlated with Dub 3 expression. In lanes 1 and 2 Dub3 is high, but Snail 1 is low; in lane 12, Snail1 is high but Dub3 is low (the opposite from lanes 1 and 2). In lane 8 there is some Snail1 expression but virtually no Dub3. This all shows lack of correlation between the two proteins in human tumors.

3. Colocalization experiment in Fig. 1c. Green Snail1 and red Dub3 should give combined yellow nuclear staining in lower right panel. Maybe one cell shows this. The results are not illustrative and do not support the authors' conclusions. However, the results with co-transfection of Snail1 and Dub3 in MDA-MB231 cells do show co-localization. The results suggest that perhaps colocalization is dependent on cell type.

4. While, as in Figure 3a, Dub3 is seen to stabilize snail1 in HEK cells, the results in Fig 3b with MDA-MB-231 cells does not follow this pattern. Snail1 drops dramatically after 0.5 hr whether Sh-anti-Dub3 RNA is present or not. It is not clear that the densitometric results are statistically significantly different from one another at least at 0.5 hr.

5. The authors show convincing evidence, in Figs 4, of the central role of Dub3 in inducing EMT and cell migration and invasion. However, these experiments do not prove that stabilization of Snail1 is the cause. Rather, the results presented in Figs 5a-b showing that Snail1 rescue of the pattern of mesenchymal protein expression is strong evidence that Dub3 works through Snail1 in producing the mesenchymal phenotype.

6. The in vivo studies in Figs. 6 clearly indicate that knockdown of Dub3 in MDA-MB-231 breast cancer cells dramatically reduces metastasis. In addition, in the DOX-induced construct breast cells (Figs 7), in the recurrence model, they demonstrate that blockade of Dub3 expression significantly reduces tumor recurrence post-operatively, an important result.

7. This reviewer was asked to focus on the molecular modeling study by the authors on the binding of WP1130 to the modeled active site of Dub3. A valid review cannot be performed since the authors do not describe how they constructed the active site of Dub3 from the active site of the UCH domain of USP2 whose x-ray crystal structure was recently elucidated. The authors do not describe the programs employed, the potential functions used and how they arrived at their proposed structure for the inhibitor Wp1130 molecule. Did the authors perform energy minimizations with or without coordinate constraints? How did the authors "construct" the

structures for the Dub3 active site where there are amino acid substitutions or deletions? How did they construct the structure for WP1130? In view of the absence of these considerations, the authors may wish to delete this study or to provide a full description of their modeling procedures.

8.To demonstrate that WP1130 binds to Dub3, the authors show that this inhibitor lowers the Dub3 melting temperature in a dose-dependent manner. These results may merely indicate that WP1130 may denature the protein as do small molecules like urea and guanidine hydrochloride and do not allow for measuring an "affinity" of Wp1130 for Dub3. Binding and affinities can be measured by spectral shifts at 280 nm or fluorescence changes. Without such data, specific binding of WP1130 to Dub3 has not been established in this study. The authors do establish that this inhibitor does reduce mammosphere formation, tumor cell migration and invasion but not via direct binding of this inhibitor to Dub3.

9.The authors conclude that agents such as Wp1130 may be effective in blocking tumor metastasis and present results shown in Figs 7e-j as proof of principle. While it is clear that inhibitors such as WP1130 are effective in reducing tumor metastasis, their effect of the viability and growth of normal cells has not been considered.

Overall, this is an important paper and should be published. However, it should be revised along the lines implied in the comments in points 1-5 and 7-9.

Point-By-Point Response to the Reviewers' Comments

We are pleased that the reviewers found our study to be novel and important, and also appreciative of their helpful and constructive comments. We have taken the comments from all reviewers seriously and revised our manuscript extensively. We believe that with our new data, the reversion has dramatically strengthened our study and addressed the reviewers' concerns. Below, we respond to the comments made by each reviewer.

Reviewer #1

1. *The authors provide promising evidence in Figure 4 that Dub3 regulates EMT, migration and invasion. However, it is unclear whether such effects are dependent on its deubiquitinating activity.*

Response: We appreciate the insightful comment from Reviewer#1. To confirm that Dub3 regulation of EMT is dependent on its deubiquitinating activity, we performed the same experiments using a catalytic-inactive Dub3 mutant (CS-Dub3). We found that only WT-Dub3 but not CS-Dub3 increased EMT, migration and invasion (Fig 4 and Supplemental Fig 4). These data indicate that the functional effect of Dub3 is dependent on its deubiquitinating activity.

2. *DUB3 inhibition is shown to inhibit breast cancer metastasis. However, it is unclear whether DUB3 overexpression promotes breast cancer metastasis. If so, is its DUB activity required for this process?*

Response: We appreciate the constructive comment from Reviewer#1. We overexpressed WT-Dub3 and CS-Dub3 mutant in MCF7 and T47D cells and found that expression of WT-Dub3 significantly induced EMT (Fig 4a-4d), increased breast cancer cells invasion and migration (Fig 4e-4f), whereas CS-Dub3 did not have these effects. These data suggest that overexpression of Dub3 can promote breast cancer cell migration and invasion and that the enzymatic activity of Dub3 is required for these effects.

3. *In Fig 3F, the authors claimed that beta-TRCP promoted Snail degradation. However, the reviewer found no such effect by beta-TRCP (lane 1 & Lane 7). The authors need to repeat the experiment in order to draw the precise conclusion.*

Response: We thank Reviewer#1 for this constructive comment. We have repeated this experiment and our results showed that overexpression of β -TRCP induced Snail1 degradation (lane 4 vs lane 1; Fig 3f), which is consistent with previous publications^{1,2}.

4. *Using computer modeling and in vitro binding assay, DUB inhibitor WP1130 was found to bind to Dub3. Subsequent experiments revealed that WP1130 inhibited IL-6-mediated Snail stability, cell migration, tumor sphere formation and primary tumor formation. Based on these results, the authors concluded that WP1130 displayed such effect likely through Dub3 inhibition. However, there is no direct evidence that WP1130 inhibits Dub3 activity. Since WP1130 also inhibits other DUBs, it is unclear whether WP1130 indeed acts through Dub3 inhibition to impact on those phenotypes. To make such claims, the authors may examine these phenotypes in the context of Dub3 deficiency.*

Response: We greatly appreciate this illuminating comment from Reviewer#1. We have performed the experiments as suggested. We knocked down Dub3 expression in MDA-MB231 and MDA-MB157 cells and treated these Dub3 deficient cells with or without WP1130 (Fig 7d-7f, Supplemental Fig 7e-7g). Consistent with standing data, WP1130 treatment significantly inhibited migration, invasion and tumorsphere-formation in MDA-MB231 and MDA-MB157 cells. However, Dub3-knockdown greatly reduced the suppressive effects mediated by WP1130. Together, these data indicate that the suppressive effects of WP1130 are mainly mediated through Dub3 inhibition.

Reviewer #2

MAJOR POINTS

1. *Regarding the Abstract I'd suggest revising the sentence "knockdown of Dub3 resulted in Snail1 destabilization, suppressed EMT, and decreased tumor initiation and metastasis. These effects were rescued by ectopic Snail1 expression". This is not accurate in regards to the results shown herein since Dub3 knockdown might revert EMT and decreases metastasis whereas Dub3 inhibition by WP1130 decreases tumor initiation. Snail1 expression rescues lung colonization but not primary tumor growth.*

Response: We greatly appreciate the insightful comment from Reviewer#2. We have revised the abstract as suggested in our revised manuscript.

2. The data involving *Drosophila* development while interesting might dilute the message. Do the authors consider Dub3 is the unique DUB stabilizing Snail1, regardless of the context?

Response: We appreciate the constructive comment from Reviewer#2. Dub3 is a conserved gene from *Drosophila* to human; knockout of Dub3 expression in *Drosophila* reduced the level of Snail1 and re-stored the expression of target genes for Snail1 (Fig 1h). These data suggest that the function of Dub3 in stabilizing Snail1 is conserved. At this moment, we cannot conclude that Dub3 is the unique DUB in stabilizing Snail1 without further systematic analyses, as other DUBs may exist that impact the stabilization of Snail1 in different cellular contexts. However, we only want to convey a message that the function of Dub3 in stabilizing Snail1 is likely conserved from *Drosophila* to mammals. We agree with Reviewer#2 that this data in *Drosophila* is only supportive but not conclusive, and we will be happy to move this data to a Supplementary Figure if Reviewer#2 prefers.

3. In Figure 4 the IF images should be improved. Same cell confluence should be shown to properly address the EMT phenotype. Moreover, better image quality regarding E-cadherin subcellular localization is necessary since that shown does not localize properly, at least in MCF7 cells. The same accounts for Fig. 5b. E-cadherin localization is not adequate in shDub3 cells since it even seems nuclear. Besides, to properly conclude in line 200 "Together, these data indicate that Dub3 can induce EMT (luminal to basal like phenotype conversion) by stabilizing Snail1 in breast cancer cells" the authors would need to silence Snail in cells MCF7 and T47D that overexpress Dub3. This way they would be able to conclude that all the effects.

Response: We highly appreciate this illuminating comment from Reviewer#2. We have performed the experiments as suggested. We also included a rescue experiment in which Snail1 was knocked down in MCF7 and T47D cells that overexpressed Dub3. We found that Snail1-knockdown reversed the EMT phenotype, and inhibited cell migration and invasion mediated by Dub3 (Fig 4 and Supplemental Fig 4). These additional data provide further support to our conclusion.

4. I'd strongly recommend not using shDub3 selected clones and repeat the experiments shown in Figure 5 with pools of cells recently infected and silenced for Dub3. Moreover, the levels of Snail in MD231 shNTC cells in Fig. 5a are extremely low compared to other panels.

Response: We apologize that we did not make a better description in our initial submission. In Figure 5, shDub3-1 and shDub3-2 are not two individual clones selected from cells with shDub3 expression. Instead, they are pool of cells with expression of two individual shDub3 (shDub3-1 and shDub3-2) that target two different regions of Dub3 mRNA. We have added this information in the Result section and the corresponding figure legend for clarity.

The low level of Snail1 in MDA-MB231 cells was due to the short-exposure of Western blot, we have replaced it with a blot using a longer-exposure in our revised manuscript (Fig 5a).

5. Regarding stemness it is surprising to find such a huge proportion of CSC within breast cancer basal cells (around 70%). I'd suggest further reviewing the literature cited and discuss their results since probably this population is fibroblastic cells and not CSC. To state that, further experiments are required, such as limiting dilution tumorigenicity assays, etc. I am also little convinced about the mammosphere forming ability shown in Fig. 5. In order to be able to conclude that, serial passaging of mammospheres should be performed to assess self-renewal ability.

Response: We highly appreciate the insightful comment from Reviewer#2. Currently, no specific markers are available for identifying breast cancer stem cells (CSCs) due to different lineages of stem or progenitor cells and the confounding effect of cellular de-differentiation/plasticity of tumor cells³. Several cell-surface markers, including CD44^{high}/CD24^{low}, CD49f, EpCAM, CD10, and CD133, have been used to enrich breast CSCs. Among them, the CD44^{high}/CD24^{low} population is commonly used and found to enrich breast CSCs by many laboratories, including those from Drs. Robert Weinberg and Kornelia Polyak⁴⁻⁹. We also found that CD44^{high}/CD24^{low} cell population is rich in CSC-like properties^{10, 11}. Based on gene expression profile¹², basal-like breast cancer (BLBC) cells are postulated to originate from breast stem/progenitor cells¹³, which reside with basal/myoepithelial cells (termed basal-like cells) of the normal breast; therefore it is not surprising that BLBC cells exhibit mesenchymal/fibroblastic characteristics and are enriched in the CD44^{high}/CD24^{low} population. Consistent with this notion, BLBC cell lines contain a high, whereas luminal breast cancer cell lines possess a low, percentage of CD44^{high}/CD24^{low} cells^{7, 14-16}. For example, Sheridan et al¹⁶, Marotta et al⁷, Ma et al¹⁵, and Su et al⁹ have shown that about 80% of MDA-MB231 cells are in the CD44^{high}/CD24^{low} population. Our results are thus consistent with

these publications.

We did consider the comment from Reviewer#2 seriously, and used a second set of breast CSC markers (CD49^{high}/CD24^{low})¹⁷⁻²⁰, to repeat our experiments. Similar to the result above, Dub3-knockdown reduced the population of CD49^{high}/CD24^{low} cells in MDA-MB231 and MDA-MB157 cells (lower panel in Fig 5g and Supplemental Fig 5h). Together, using two sets of cell-surface markers, we found that Dub3 is critical in regulating CSC-like properties by stabilizing Snail1.

We also performed mammosphere-formation assay with serial passage, as suggested by Reviewer#2. We found that Dub3-knockdown greatly reduced the number and size of both primary and secondary mammospheres in MDA-MB231 and MDA-MB157 cells (Fig 5f).

6. In regards to WP1130 results, it is necessary to present data concerning the cytotoxicity assays it is published that that WP1130 has pro-apoptotic activity and those data are necessary if the authors suggest its use as possible cancer treatment.

Response: We greatly appreciate the insightful comment from Reviewer#2. WP1130 has been shown to induce apoptosis in leukemia, multiple myeloma, and lymphoma cells²¹⁻²³. Although this drug is less effective in solid tumors, combined treatment of WP1130 promotes tumor cell sensitivity to doxorubicin²⁴, the Bcl2-family inhibitors (ABT263, ABT737)^{25, 26}, and radiation²⁷ in hepatoma, glioblastoma, and lung cancers, respectively. To examine cytotoxicity of WP1130 in breast cancer, we treated breast tumor cells (MDA-MB231) and immortalized normal human breast epithelial cells (MCF10A) with different doses of WP1130 (0.5 and 1 μ M) for different time intervals (24 and 48 hrs). With concentration up to 1 μ M WP1130 for 48 hr, we observed no apoptosis in these cells (Supplemental Fig 7d). In our analyses, we used 0.5 μ M WP1130 for 24 and 4 hr to measure migration and invasion, respectively, thus the effects observed are likely not due to any cytotoxicity of this compound.

7. The results regarding Snail1 IHC are intriguing. Snail1 IHC in tumor samples has remained elusive for many years and in many laboratories so it'd be convincing to show several images of different samples showing Snail1 by IHC. Besides, the authors are vague about these tumor samples and they should clarify which kind of breast tumor samples are these. Coexpression would be expected in TNBC but not in all breast tumor samples according to the authors' hypothesis. It might be worth to show Snail and Dub3 association to different types of breast tumor types. Moreover, Fig. S8 is not clear at all. And its Legend refers to expression of both Snail and Dub3 which is not mentioned within the text.

Response: We highly appreciate the constructive comment from Reviewer#2. We have performed IHC analyses using a breast TMA generated by the Bio-specimen Repository in our NCI-designed Markey Cancer Center at the University of Kentucky College of Medicine. This TMA contains 334 cases of breast tumor specimens, including 110 luminal, 59 HER2-overexpressing, and 165 triple-negative breast cancer (Tables 1-3 in our revised manuscript). We found that high expression of both Dub3 and Snail1 are associated with TNBC. We also provide three different tumor sample IHC images, both positive and negative of Snail1 and Dub3 (Fig 8a), in our revised manuscript as suggested by Reviewer#2. In addition, we have included a more detailed description on these IHC analyses in the sections of Results and Methods, as well as in the corresponding figure legend.

We apologize that we did not provide detailed description on Fig. S8. Based on gene expression profiles (datasets from Finak and Esserman in Oncomine), high level of Dub3 mRNA was found in the invasive ductal breast carcinoma compared to normal breast tissue. We have provided more description in the figure legend of Fig. S8 in our revised manuscript.

MINOR POINTS

1. Line 88: it doesn't seem adequate to state "To understand the regulation of Snail1 in breast cancer, we purified the Snail1 complexes from nuclear extracts of 20 liters HeLa" since HeLa are not breast cancer cells. I'd suggest rephrasing.

Response: We appreciate the constructive comment from Reviewer#2. We have rephrased these statements in our revised manuscript (please see line 2 in page 5).

2. Figure 2c: the authors should mention why there are two clear bands when N-terminal Snail1 is cloned; whereas the results regarding binding to the C-terminal part of Dub3 are not conclusive due to the low expression of this construct.

Response: We appreciate the constructive comment from Reviewer#2. The C-terminal of Snail1 contains four Zinc-finger motifs that are required for interaction with DNA. In contrast, the N-terminal Snail1 contains multiple Serine and Threonine residues and is subjected to hyper-phosphorylation as shown by multiple laboratories,

including us^{1, 2, 28}. We speculate that the upper band of N-terminal Snail1 (Fig 2c) is the phosphorylated state of Snail1. Both Co-IP and GST pull-down experiments (Fig 2c and 2d) indicate that the N-terminal Snail1 interacts with Dub3, and suggest, although do not definitely prove, that the interaction of N-terminal Snail1 with Dub3 is sequence specific and is not dependent on the phosphorylation status of Snail1. Future systematic investigations are warrant to unveil their interesting interaction.

We also appreciate the insightful comment from Reviewer#2 regarding the low abundance of C-terminal Dub3 (Fig 3e). The C-terminal fragment of Dub3 only contains 128 amino acids (about 14 kDa in molecular weight), and we speculate that this may due to the longer transfer timer for the western blot; small molecular-weight molecules may have passed through the nitrocellulose membrane. We repeated this experiment with a shorter transfer time for the western blot. Our results showed that the C-terminal Dub3 was not expressed in low levels and that Snail1 did not interact with the C-terminal Dub3 (Fig 2e).

3. Line 138: Do the authors really did this: "We also purified full-length Dub3 from a GST-Dub3 fusion protein by cleaving and removing the GST portion with the tobacco etch virus (TEV) protease".

Response: We appreciate this constructive comment from Reviewer#2. We rephrased our revised manuscript to clarify this procedure (please see line 19 in page 6).

4. Line 173: the references 32 and 33 do not seem adequate herein.

Response: We appreciate this constructive comment from Reviewer#2. We have removed these two references and include the appropriate references.

5. Line 207: the sentence is not adequate, Dub3 knockdown doesn't restore E-cadherin levels, it increases them.

Response: We appreciate this constructive comment from Reviewer#2. We have corrected it in our revised manuscript (please see line 1 from bottom in page 8).

6. Line 287: S7b instead of S7c.

Response: We appreciate this constructive comment from Reviewer#2. We have changed it in our revised manuscript (please see line 9 from bottom in page 11).

Reviewer #3

1. The initial experiments, the results of which are described in Figures 1 and 2, are poorly defined and give results that even contradict the contention of the authors and the results presented further on. The protocols for Figures 1 are undefined. Are the authors transfecting constructs for FLAG-Snail1 and HA-Dub3 into HEK 239 cells and then immunoprecipitating each? If, as the authors document further on, CS-Dub3 is inactive, why is there a rather high level of Snail1 in Fig 1A? Also, how is the intracellular concentration of Dub 3 increased in Fig 1B? Is its expression under the control of an inducer? If so what is the construct? This has to be defined.

Response: We apologize that we did not describe these experiments in better detail. In Fig 1A, Flag-Snail1 was co-expressed with either vector (lanes 1 and 2) or HA-Dub3 in HEK293 cells. After 42 hr of transfection, cells from lane 2 were treated with the proteasome inhibitor MG132 for 6 hr. Then, 48 hr post-transfection, all cells were harvested and lysates were prepared for Western blot analysis (not immunoprecipitation). Because Snail1 is a labile protein and subjected to constant protein ubiquitination and degradation^{1, 2}, treatment with MG132 (lane 2) served as a positive control, demonstrating a stable Snail1 protein; whereas the sample in lane 1 (with no MG132 treatment) served as an additional control. Similar to lane 2, co-expression of WT-Dub3 (lane 3) induced Snail1 stabilization (lane 3 vs lane 2). However, expression of catalytic-inactive Dub3 (CS-Dub3; lane 4) did not have this effect, because results obtained from cells expressing Snail1 with no MG132 treatment and cells expressing Snail1 with inactive Dub3 (lane 1 vs lane 4) are similar. These data indicate that Dub3 can stabilize Snail1 and that the catalytic activity of Dub3 is required for this effect.

In Fig 2B, we co-expressed a fixed amount of Flag-Snail1 (0.5 μ g) with increasing concentrations of wild-type HA-Dub3 (0.5, 1, and 2 μ g in lanes 2, 3, and 4, respectively) in HEK293 cells. Then, 48 hr post-transfection, all cells were harvested and lysates were prepared for Western blot analysis (not immunoprecipitation) to examine the expression of HA-Snail1 and Flag-Dub3. These data indicate that Dub3 can stabilize Snail1 in a dose-dependent manner.

We are sorry that we did not make these experimental details clear in our original submission, we have provided experimental details in the Methods section and the corresponding figure legend.

2. Fig1e is supposed to show that in a variety of human tumors Snail1 expression correlated with Dub 3 expression. In lanes 1 and 2 Dub3 is high, but Snail 1 is low; in lane 12, Snail1 is high but Dub3 is low (the opposite from lanes 1 and2). In lane 8 there is some Snail1 expression but virtually no Dub3. This all shows lack of correlation between the two proteins in human tumors.

Response: We appreciate the insightful comment from Reviewer#3. Unlike cancer cell lines, patient tumor samples are heterogeneous, containing stromal cells and infiltrated lymphocytes. It is difficult to isolate tumor cells from fresh tumor samples for Western blot analysis in a short time-frame without compromising the protein stability of Snail1. Although the levels of Dub3 and Snail1 are not perfectly matched or even as well as that seen in cancer cell lines (Fig 1D), the level of Dub3 is positively correlated with Snail1 in the majority of fresh tumor samples. In addition, we performed IHC analysis for the expression of Dub3 and Snail1 in 334 formalin-fixed and paraffin-embedded breast tumor samples (TMA samples), and found that the expression of Dub3 positively correlated with Snail1 in triple-negative breast cancer samples (Tables 1-3 and Fig 8). Taken together, these systematic analyses ranging from cancer cell lines, fresh tumor samples and IHC analyses, indicate that Dub3 expression positively correlates with the level of Snail1 in breast tumor.

3. Colocalization experiment in Fig. 1c. Green Snail1 and red Dub3 should give combined yellow nuclear staining in lower right panel. Maybe one cell shows this. The results are not illustrative and do not support the authors' conclusions. However, the results with co-transfection of Snail1 and Dub3 in MDA-MB231 cells do show colocalization. The results suggest that perhaps colocalization is dependent on cell type.

Response: We apologized that we did not describe this experiment more clearly in the original submission. In Fig 1c, Snail1 and Dub3 were co-expressed in HEK293 cells. Then, 48 hr post-transfection, cells were fixed and subjected to immunofluorescent stain for the expression of Snail1 (green), Dub3 (red), and nuclei (with DAPI; blue). The three staining patterns were merged, and shown in the lower right panel of Fig 1c in our original submission. The three-color combinations may generate hues ranging from pink to white, depending on the intensities of the three stains. To avoid the confusion of a three-color combination, we removed the DAPI staining (nuclei; blue) from the merge panel in our revised manuscript (lower right panel of Fig 1c). Now it becomes clear that a yellow color is generated in the merged panel, which indicates that Dub3 and Snail1 co-localize in the nucleus. Of note, high Dub3 expression was observed along with a high intensity of Snail1 expression in 4 individual cells at the top right corner in each panel. However, one cell (with white arrow) had a low intensity for Snail1 staining, because this cell did not show Dub3 expression (no red color). Together, these data indicate that Dub3 expression stabilizes Snail1 in the nucleus. This result is consistent with the data in MDA-MB231 cells (Fig 2h) and indicates that the co-localization of Dub3 and Snail1 is not cell-type dependent.

4. While, as in Figure 3a, Dub3 is seen to stabilize snail1 in HEK cells, the results in Fig 3b with MDA-MB-231 cells does not follow this pattern. Snail1 drops dramatically after 0.5 hr whether Sh-anti-Dub3 RNA is present or not. It is not clear that the densitometric results are statistically significantly different from one another at least at 0.5 hour.

Response: We apologize that we did not provide greater detail in the description of this experiment. The protein level in each cell is highly dynamic, balanced by protein synthesis and protein degradation. Fig 3a and 3b utilize the classic method for measuring protein stability²⁹. In this method, cells were treated with Cycloheximide (CHX), an inhibitor of protein biosynthesis, for different time intervals (0, 15, 30, 60, 120, 240 minute in Fig 3A). Cells were harvested and lysates prepared to assess the expression of Snail1 by Western blot analysis. Because protein biosynthesis is blocked by CHX, the decrease in Snail1 level at each time point indicates the rate of Snail1 degradation. In Fig 3a, the level of Snail1 in cells expressing both Snail1 and Dub3 became stabilized when compared with cells expressing Snail1 and vector (right half vs left half; Fig 3a). This is because Dub3 can stabilize Snail1 by removing the ubiquitination from Snail1. In Fig 3b, knockdown of Dub3 greatly enhanced the degradation of Snail1 in MDA-MB231 cells.

These experiments were repeated as three independent measures, and the densitometric graphs represent the average of these three independent experiments. We apologize for not making the experimental design of these experiments clear in the initial submission, and we now include additional experimental details in the Methods section.

5. The authors show convincing evidence, in Figs 4, of the central role of Dub3 in inducing EMT and cell migration and invasion. However, these experiments do not prove that stabilization of Snail1 is the cause. Rather, the results presented in Figs 5a-b showing that Snail1 rescue of the pattern of mesenchymal protein expression is strong evidence that Dub3 works through Snail1 in producing the mesenchymal phenotype.

Response: We appreciate the constructive comment from Reviewer# 3. This question is the similar to the comment#3 from Reviewer#2. Please see our detailed response to this comment. Briefly, we have included a rescue experiment in Fig 4, in which we knocked down Snail1 expression in MCF7 and T47D cells overexpressing Dub3. We found that Snail1-knockdown significantly blocked the increased migration, invasion, and CSC-like properties mediated by Dub3. These data support our conclusion that the function of Dub3 is mainly mediated through Snail1 in EMT.

6. *The in vivo studies in Figs. 6 clearly indicate that knockdown of Dub3 in MDA-MB-231 breast cancer cells dramatically reduces metastasis. In addition, in the DOX-induced construct breast cells (Figs 7), in the recurrence model, they demonstrate that blockade of Dub3 expression significantly reduces tumor recurrence post-operatively, an important result.*

Response: We appreciate the positive comment from Reviewer# 3.

7. *This reviewer was asked to focus on the molecular modeling study by the authors on the binding of WP1130 to the modeled active site of Dub3. A valid review cannot be performed since the authors do not describe how they constructed the active site of Dub3 from the active site of the UCH domain of USP2 whose x-ray crystal structure was recently elucidated. The authors do not describe the programs employed, the potential functions used and how they arrived at their proposed structure for the inhibitor Wp1130 molecule. Did the authors perform energy minimizations with or without coordinate constraints? How did the authors "construct" the structures for the Dub3 active site where there are amino acid substitutions or deletions? How did they construct the structure for WP1130? In view of the absence of these considerations, the authors may wish to delete this study or to provide a full description of their modeling procedures.*

Response: We appreciate the insightful comment from Reviewer#3. A comparative model structure construction of the Dub3-UCH domain and docking studies with WP1130 had been described in the 'Methods' section. In brief, the programs MODELLER and PubChem were used for constructing the protein and the compound structures, respectively, and the SwissDock program was used for docking studies. The USP2-UCH and Dub3-UCH domains share about 50% sequence identity, including the catalytic residues at the active site. The core catalytic residues, namely the Cys, His and Asp triads, are strictly conserved among the USP family members³⁰, including USP2 and Dub3 (USP17) used in the current studies. The SwissDock program adopts the CHARMM simulation program which employs numerous conformational sampling methods and free energy minimization. Several top solutions from the docking studies display similar binding modes at the active site (all interacting with the same backbone atoms with the similar orientations) and only the top solution was chosen for presentation in the manuscript. In this revised manuscript, references to the Method section (please see 'Methods' for details) and the program MODELLER were added.

8. *To demonstrate that WP1130 binds to Dub3, the authors show that this inhibitor lowers the Dub3 melting temperature in a dose-dependent manner. These results may merely indicate that WP1130 may denature the protein as do small molecules like urea and guanidine hydrochloride and do not allow for measuring an "affinity" of Wp1130 for Dub3. Binding and affinities can be measured by spectral shifts at 280 nm or fluorescence changes. Without such data, specific binding of WP1130 to Dub3 has not been established in this study. The authors do establish that this inhibitor does reduce mammosphere formation, tumor cell migration and invasion but not via direct binding of this inhibitor to Dub3.*

Response: We appreciate the insightful comment from Reviewer#3. Negative T_m shifts could be attributed to the compound destabilizing the protein or the compound aggregating and causing early destabilization³¹. However, these types of negative shifts (ΔT_m) were observed for the compounds which contain heavy metal atoms, such as bromine (Br) in WP1130, and generate energetically unfavorable strains when interacting with the proteins^{32, 33}. In addition, although binding affinity and the related thermodynamic parameters were not measured in the current studies, a direct binding between Dub3-UCH and WP1130 was demonstrated by the shifts during the EMSA analysis (supplementary Fig 7b). In the revised manuscript, an explanation of the negative T_m shift and the pertaining references were included in the main text.

In addition, we knocked down Dub3 expression in MDA-MB2231 and MDA-MB157 cells, knockdown of Dub3 expression greatly reduced the suppressive effects on migration, invasion and tumorsphere-formation mediated by WP1130 (Fig 7d-7f, and Supplemental Fig 7e-7g), indicating that the suppressive function of WP1130 is likely mediated through Dub3.

9. *The authors conclude that agents such as WP1130 may be effective in blocking tumor metastasis and present*

results shown in Figs 7e-j as proof of principle. While it is clear that inhibitors such as WP1130 are effective in reducing tumor metastasis, their effect of the viability and growth of normal cells has not been considered.

Response: We appreciate the insightful comment from Reviewer#3. We have included cytotoxicity assessment of WP1130 on a normal human breast epithelial cells line (MCF10A) and MDA-MB231 cells with different doses of WP1130 (0.5 and 1 μ M) at different time intervals (24 and 48 hr). With concentration up to 1 μ M WP1130 for 48 hr, we observed no apparent cytotoxicity in these cells (Supplemental Fig 7d). In our study, we treated cells with 0.5 μ M WP1130 for 4-24 hr to assess functional changes, thus the effects observed are likely not due to cytotoxicity of this compound.

REFERENCES

1. Yook, J.I., Li, X.Y., Ota, I., Fearon, E.R. & Weiss, S.J. Wnt-dependent regulation of the E-cadherin repressor snail. *The Journal of biological chemistry* **280**, 11740-11748 (2005).
2. Zhou, B.P. *et al.* Dual regulation of Snail by GSK-3 β -mediated phosphorylation in control of epithelial-mesenchymal transition. *Nature cell biology* **6**, 931-940 (2004).
3. Visvader, J.E. & Lindeman, G.J. Cancer stem cells: current status and evolving complexities. *Cell stem cell* **10**, 717-728 (2012).
4. Al-Hajj, M., Wicha, M.S., Benito-Hernandez, A., Morrison, S.J. & Clarke, M.F. Prospective identification of tumorigenic breast cancer cells. *Proceedings of the National Academy of Sciences of the United States of America* **100**, 3983-3988 (2003).
5. Cordenonsi, M. *et al.* The Hippo transducer TAZ confers cancer stem cell-related traits on breast cancer cells. *Cell* **147**, 759-772 (2011).
6. Mani, S.A. *et al.* The epithelial-mesenchymal transition generates cells with properties of stem cells. *Cell* **133**, 704-715 (2008).
7. Marotta, L.L. *et al.* The JAK2/STAT3 signaling pathway is required for growth of CD44CD24 stem cell-like breast cancer cells in human tumors. *The Journal of clinical investigation* **121**, 2723-2735 (2011).
8. Pattabiraman, D.R. *et al.* Activation of PKA leads to mesenchymal-to-epithelial transition and loss of tumor-initiating ability. *Science* **351**, aad3680 (2016).
9. Su, Y. *et al.* Somatic Cell Fusions Reveal Extensive Heterogeneity in Basal-like Breast Cancer. *Cell reports* **11**, 1549-1563 (2015).
10. Dong, C. *et al.* Loss of FBP1 by Snail-mediated repression provides metabolic advantages in basal-like breast cancer. *Cancer cell* **23**, 316-331 (2013).
11. Wu, Y. *et al.* The Deubiquitinase USP28 Stabilizes LSD1 and Confers Stem-Cell-like Traits to Breast Cancer Cells. *Cell reports* (2013).
12. Prat, A. *et al.* Phenotypic and molecular characterization of the claudin-low intrinsic subtype of breast cancer. *Breast cancer research : BCR* **12**, R68 (2010).
13. Visvader, J.E. Keeping abreast of the mammary epithelial hierarchy and breast tumorigenesis. *Genes & development* **23**, 2563-2577 (2009).
14. Blick, T. *et al.* Epithelial mesenchymal transition traits in human breast cancer cell lines. *Clinical & experimental metastasis* **25**, 629-642 (2008).
15. Ma, F. *et al.* Enriched CD44(+)/CD24(-) population drives the aggressive phenotypes presented in triple-negative breast cancer (TNBC). *Cancer letters* **353**, 153-159 (2014).
16. Sheridan, C. *et al.* CD44+/CD24- breast cancer cells exhibit enhanced invasive properties: an early step necessary for metastasis. *Breast cancer research : BCR* **8**, R59 (2006).
17. Lo, P.K. *et al.* CD49f and CD61 identify Her2/neu-induced mammary tumor-initiating cells that are potentially derived from luminal progenitors and maintained by the integrin-TGF β signaling. *Oncogene* **31**, 2614-2626 (2012).
18. Meyer, M.J. *et al.* CD44posCD49fhiCD133/2hi defines xenograft-initiating cells in estrogen receptor-negative breast cancer. *Cancer research* **70**, 4624-4633 (2010).
19. To, K. *et al.* Y-box binding protein-1 induces the expression of CD44 and CD49f leading to enhanced self-renewal, mammosphere growth, and drug resistance. *Cancer research* **70**, 2840-2851 (2010).
20. Yin, Y. *et al.* CD151 represses mammary gland development by maintaining the niches of progenitor cells. *Cell cycle* **13**, 2707-2722 (2014).
21. Bartholomeusz, G.A. *et al.* Activation of a novel Bcr/Abl destruction pathway by WP1130 induces apoptosis of chronic myelogenous leukemia cells. *Blood* **109**, 3470-3478 (2007).

22. Kapuria, V. *et al.* Deubiquitinase inhibition by small-molecule WP1130 triggers aggresome formation and tumor cell apoptosis. *Cancer research* **70**, 9265-9276 (2010).
23. Sun, H. *et al.* Bcr-Abl ubiquitination and Usp9x inhibition block kinase signaling and promote CML cell apoptosis. *Blood* **117**, 3151-3162 (2011).
24. Liu, H. *et al.* WP1130 increases doxorubicin sensitivity in hepatocellular carcinoma cells through usp9x-dependent p53 degradation. *Cancer letters* **361**, 218-225 (2015).
25. Karpel-Massler, G. *et al.* Inhibition of deubiquitinases primes glioblastoma cells to apoptosis in vitro and in vivo. *Oncotarget* **7**, 12791-12805 (2016).
26. Peddaboina, C. *et al.* The downregulation of Mcl-1 via USP9X inhibition sensitizes solid tumors to Bcl-xl inhibition. *BMC cancer* **12**, 541 (2012).
27. Kushwaha, D. *et al.* USP9X inhibition promotes radiation-induced apoptosis in non-small cell lung cancer cells expressing mid-to-high MCL1. *Cancer Biol Ther* **16**, 392-401 (2015).
28. Dominguez, D. *et al.* Phosphorylation regulates the subcellular location and activity of the snail transcriptional repressor. *Molecular and cellular biology* **23**, 5078-5089 (2003).
29. Zhou, P. Determining protein half-lives. *Methods Mol Biol* **284**, 67-77 (2004).
30. Zhang, W. *et al.* Contribution of active site residues to substrate hydrolysis by USP2: insights into catalysis by ubiquitin specific proteases. *Biochemistry* **50**, 4775-4785 (2011).
31. Cummings, M.D., Farnum, M.A. & Nelen, M.I. Universal screening methods and applications of ThermoFluor. *J Biomol Screen* **11**, 854-863 (2006).
32. McDonnell, P.A. *et al.* Assessing compound binding to the Eg5 motor domain using a thermal shift assay. *Anal Biochem* **392**, 59-69 (2009).
33. Silvestre, H.L., Blundell, T.L., Abell, C. & Ciulli, A. Integrated biophysical approach to fragment screening and validation for fragment-based lead discovery. *Proceedings of the National Academy of Sciences of the United States of America* **110**, 12984-12989 (2013).

REVIEWERS' COMMENTS:

Reviewer #1 (Remarks to the Author):

The authors have adequately addressed all of my concerns. The study reveals the novel role of DUB3 in EMT and cancer metastasis through promoting Snail stability. The study is novel and well supported by experimental data. It is now suitable for publications at Nature Communications.

Reviewer #2 (Remarks to the Author):

Dub3 Inhibition Suppresses Breast Cancer Invasion and Metastasis by Promoting Snail1

The manuscript by Wu and coauthors has clearly improved upon revision. The authors have adequately answered to the reviewers' comments, performing the necessary experiments to strengthen their conclusions. It is now much clearer the role played by Dub3 in the stabilization of Snail in basal breast cancer cell lines and the consequences of such regulation. The authors also illustrate the benefits of targeting Dub3 in such setting, addressing its importance as a potential therapeutic target.

I'd convincingly recommend the manuscript to be published without major changes.

Minor points

I still consider that the Drosophila results fit better in the supplementary data but I leave this decision to the authors' choice.

Lines 338-339: I'd rephrase this sentence. Actually, the tight correlation between Dub3 and Snail in samples or cell lines does not confirm the regulation of Snail by Dub3. In any case, it might strengthen the authors' hypothesis of such regulation.

Lines 354-356: I'd downscale this affirmation. Dub3 might be one of the molecules conveying inflammation signals to Snail stabilization and so on, but there are undoubtedly other players involved.

Editorial Note: Reviewer 3 was not available to comment on the revised manuscript. Therefore we sought further feedback from a reviewer with overlapping expertise. The reviewer communicated to the editors only that the authors' responses to the concerns raised by reviewer 3 were satisfactory.

Point-By-Point Response to the Reviewers' Comments

We are pleased that Reviewer#1 and Reviewer#2 found our revised manuscript to be “novel and well supported by experimental data; and it is now suitable for publications at *Nature Communications*”. Below, we respond to the minor comments from Reviewer#2.

Reviewer #1

“The authors have adequately addressed all of my concerns. The study reveals the novel role of DUB3 in EMT and cancer metastasis through promoting Snail stability. The study is novel and well supported by experimental data. It is now suitable for publications at Nature Communications”.

Response: We appreciate the support from Reviewer#1.

Reviewer #2

Minor points

(1) I still consider that the Drosophila results fit better in the supplementary data but I leave this decision to the authors' choice.

Response: We thank the constructive comment from Reviewer#2. Dub3 is a conserved gene from *Drosophila* to human; knockout of Dub3 expression in *Drosophila* reduced the level of Snail1 and re-stored the expression of target genes for Snail1 (Fig 1h). These data suggest that the function of Dub3 in stabilizing Snail1 is conserved. This finding is consistent with our other observations found in mammalian cells, tumor samples and animal model. We prefer to keep this result in Fig 1, because this is a strong supporting data for our conclusion.

(2) Lines 338-339: I'd rephrase this sentence. Actually, the tight correlation between Dub3 and Snail in samples or cell lines does not confirm the regulation of Snail by Dub3. In any case, it might strengthen the authors' hypothesis of such regulation.

Response: We appreciate the comment from Reviewer#2. We have rephrased this sentence and changed the original wording “...confirms the regulation of Snail1 by Dub3” into “...confirms their potential regulation” in the revised manuscript (please see page 13).

(3) Lines 354-356: I'd downscale this affirmation. Dub3 might be one of the molecules conveying inflammation signals to Snail stabilization and so on, but there are undoubtedly other players involved.

Response: We appreciate the comment from Reviewer#2. We have downscaled this conclusion and changed the initial wording “...Dub3 is the long-sought missing molecule...” into “...Dub3 is one of the long-sought missing molecule...” in the revised manuscript (please see page 13).